# Active droplets through enzyme-free, dynamic phosphorylation

Simone M. Poprawa [1], Michele Stasi [1], Brigitte A. K. Kriebisch [1], Monika Wenisch [1], Judit Sastre[1] & Job Boekhoven [1] ✉

Life continuously transduces energy to perform critical functions using energy stored in reactive molecules like ATP or NADH. ATP dynamically phosphorylates active sites on proteins and thereby regulates their function. Inspired by such machinery, regulating supramolecular functions using energy stored in reactive molecules has gained traction. Enzyme-free, synthetic systems that use dynamic phosphorylation to regulate supramolecular processes have not yet been reported, to our knowledge. Here, we show an enzyme-free reaction cycle that consumes the phosphorylating agent monoamidophosphate by transiently phosphorylating histidine and histidine-containing peptides. The phosphorylated species are labile and deactivate through hydrolysis. The cycle exhibits versatility and tunability, allowing for the dynamic phosphorylation of multiple precursors with a tunable half-life. Notably, we show the resulting phosphorylated products can regulate the peptide's phase separation, leading to active droplets that require the continuous conversion of fuel to sustain. The reaction cycle will be valuable as a model for biological phosphorylation but can also offer insights into protocell formation.

In chemically fueled supramolecular chemistry, a reaction cycle is coupled to a supramolecular process[1–3]. Specifically, the reaction cycle catalytically converts fuel into waste, which regulates the supramolecular behavior of the catalyst. For such chemically fueled reaction cycles, we require a thermodynamically unstable molecule (the fuel), which can be converted into a thermodynamically more stable state (waste) in an exergonic reaction. That way, the equilibrium position for the fuel-to-waste equilibrium lies on the waste side such that most fuel will eventually be converted into waste. This fuel-to-waste conversion should be slow so that a catalyst can accelerate it. Put differently, the fuel should be thermodynamically unstable, yet kinetically inert. In the catalytic reaction cycle, the catalyst accelerates the fuel-to-waste conversion. In doing so, the catalyst is temporarily activated by reacting with the fuel, after which it spontaneously deactivates. In other words, the catalyst can undergo numerous activation-deactivation cycles, and the fuel is converted fast due to the presence of the catalyst.

From the catalyst's point of view, it is transiently activated in an endergonic reaction by reacting the fuel to waste. Adenosine triphosphate (ATP) is an excellent fuel that fulfills all of the requirements above. ATP itself is thermodynamically unstable but kinetically stable[4,5]. The hydrolysis of ATP to adenosine diphosphate (ADP) releases a relatively high amount of energy but not spontaneously; it requires ATPases (an enzyme as the catalyst) or other enzymes to liberate the energy through hydrolysis. Doing so transiently activates the ATPase through phosphorylation of its active site. This transient activation step induces a change in the ATPase, which can be used to drive supramolecular processes, such as a change in the enzyme, which pumps ions from one side to the other of a membrane[6]. Besides the canonical serine as a phosphorylated site, other amino acids can also be a target for transient phosphorylation. For example, histidine is used as a phosphate shuttle in the active site of kinases and phosphatases[7–10]. In the case of fructose-2,6-bisphosphatase, the 3-phoshohistidine is formed and broken down in the active site, to dephosphorylate the fructose-2,6-bisphosphate to form fructose-6-phosphate and inorganic phosphate[9].

Inspired by the use of fuels to regulate biological supramolecular processes, analogous fuel-consuming catalytic reaction cycles have

[1]Department of Bioscience, Technical University of Munich, Lichtenbergstrasse 4, 85748 Garching, Germany. ✉e-mail: job.boekhoven@tum.de

been devised[11–13]. For example, methylating agents are a popular fuel combined with carboxylates as catalysts to form labile, transient methyl esters[14–16]. Carbodiimides[2] are a popular fuel to activate carboxylates to form transient anhydrides[17–24], active esters[25–27], or oxazolones[28]. Because of its biological relevance, ATP is also frequently used to regulate supramolecular processes like the formation of fibers[29], or the assembly of lipids[30], or assembling DNA[31]. Some of these ATP-driven cycles use their ATP to dynamically phosphorylate a substrate by using phosphorylating (kinases) and dephosphorylating enzymes (phosphatases)[29,32,33]. Given the multiple enzymes involved, such cycles can be complicated. Their need for highly evolved enzymes also offers limited insight into the roles phosphorylation may potentially have played before life existed[34,35], even though there is great progress in phosphorylation without enzymes[36,37], but with prebiotically plausible phosphorylating agents[38–40]. So far, enzyme-free reaction cycles coupled to a supramolecular process fueled by phosphorylation agents have not yet been reported, to the best of our knowledge.

Here, we show a chemical reaction cycle that phosphorylates and dephosphorylates amino acids and peptides at the expense of simple, prebiotically plausible phosphorylating agents as fuel (Figs. 1–2). The reaction cycle is enzyme-free and produces a phosphorylated amino acid or peptide that hydrolyzes without requiring enzymes and with a tunable half-life. Finally, we demonstrate that dynamic phosphorylation can be coupled to the formation of complex-coacervate-based droplets as a model for protocells. We envision the cycle as a great, minimal model for biological dynamic phosphorylation and can also give insights into dynamic phosphorylation before the existence of highly evolved enzymes.

## Results and Discussion

The reaction cycle combines the phosphorylation of an amino acid (activation) and its subsequent dephosphorylation (deactivation), which closes the reaction cycle. Phosphorylation requires a phosphorylating agent that we refer to as fuel. Deactivation through hydrolysis can be spontaneous and sped up by catalysis. Thus, a dynamically phosphorylated amino acid appears in response to fuel, which can only be sustained by the cycle's continuous fuel consumption. We chose potassium monoamidophosphate (MAP) as a phosphorylation agent because it is known to phosphorylate imidazole[41], yet, it is relatively inert towards hydrolysis. Indeed, we found MAP's half-life of hydrolysis to be $16.5 \pm 0.2$ h by $^{31}$P nuclear magnetic resonance ($^{31}$P-NMR) in 500 mM 2-($N$-morpholino)ethane sulfonic acid (MES) buffer at pH 6.5 and room temperature (Fig. 2a, Supplementary Methods B, Supplementary Table 5 and Supplementary Fig. 1). In contrast, in the presence of 75 mM histidine, MAP's half-life for the observed hydrolysis decreased to $2.97 \pm 0.1$ h (Supplementary Table 6). Thus, histidine accelerated the conversion of 80 mM MAP, through its phosphorylation[42,43]. We tested six further amino acids with side groups that are known to be phosphorylated in biological context —arginine, lysine, cysteine, serine, tyrosine, and aspartic acid (Fig. 2b)[44,45]. We were specifically interested in amino acids because amino acids and peptides are powerful building blocks for self-assembly. For all amino acids, MAP's half-life for observed hydrolysis remained similar to the half-life in the absence of amino acids (Supplementary Table 6). Thus, none of the other amino acids observably catalyzed the fuel conversion. Indeed, no phosphorylated products were made in observable amounts by $^{31}$P-NMR (Supplementary Fig. 2). We focused on histidine as a catalyst and tested two more prebiotically relevant fuels, i.e., diamidophosphate (DAP) and trimetaphosphate (TMP, Fig. 2c and Supplementary Fig. 3)[38–40]. With 75 mM histidine, no significant amount of TMP was converted over 80 h. In contrast, DAP was converted with an observable half-life of $12 \pm 0.3$ h, which is significantly faster than without histidine at an observable half-life of $190 \pm 50$ h.

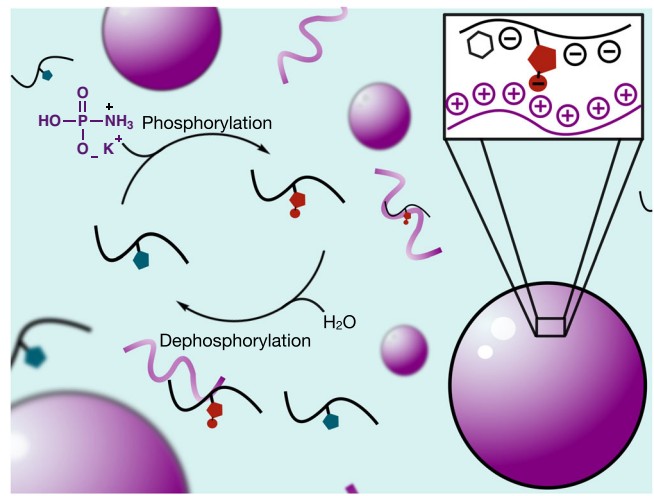

**Fig. 1 | Active coacervate-based compartments by the dynamic phosphorylation of peptides at the expense of a simple phosphorylating agent.** A schematic representation of the chemical reaction cycle coupled to the formation of complex coacervate-based droplets. The chemical reaction cycle converts a deactivated, histidine-containing precursor peptide (blue) to an activated, phosphohistidine-containing product (red) at the expense of the chemical fuel monoamidophosphate (MAP). The precursor acts as a catalyst for the hydrolysis of MAP. The activated product can interact with poly-arginine (purple) to form liquid-liquid phase-separated complex coacervate droplets. The droplets dissolve upon deactivation (dephosphorylation) of the product and the precursor can be recycled.

We tested the pathway of the histidine-catalyzed fuel consumption. By $^{31}$P-NMR, we observed the conversion of MAP, the production of inorganic phosphate ($P_i$), and the emergence and decay 1-phosphohistidine (1-pHis), 3-phosphohistidine (3-pHis), and 1,3-bisphosphohistidine (1,3-bpHis, Fig. 2d, e and Supplementary Fig. 4). Thus, all possible phosphorylated histidine species were observed. Even more exciting was that all species were only transiently present at the expense of MAP, pointing to the dynamic phosphorylation of histidine. The concentration of 3-pHis reached the highest maximum concentration of all species, which peaked at around 9 h. The others, 1-pHis and 1,3-bpHis, peaked earlier and at lower concentrations. After 60 h, all 75 mM histidine was deactivated to its dephosphorylated state, and all MAP was converted into inorganic phosphate. Besides the dynamic phosphorylation of histidine, no notable side reactions were observed. The same phosphorylated species were observed when we used DAP as a fuel. However, some side products were found. We conclude that histidine is a catalyst for the hydrolysis of MAP and DAP and is dynamically phosphorylated.

To quantitatively understand the reactions taking place in the cycle, we describe the following reactions in a kinetic model (Supplementary Methods B and Supplementary Table 5): the MAP hydrolysis to form inorganic phosphate (Fig. 2a), the reaction of MAP with histidine yielding 1-pHis or 3-pHis, the reaction of MAP with 1-pHis or 3-pHis to form 1,3-bispHis. Each of these activation reactions releases one molecule of ammonia. Additionally, we included in the model the dephosphorylation of each phosphorylated species (deactivation). A *Levenberg-Marquardt* fitting method fitted the model's rate constants to predict the experimental data (Supplementary Methods B and Supplementary Table 5). Intramolecular isomerization from 1-pHis to 3-pHis is not described in the model as it is known not to occur[46]. In contrast, intermolecular phosphoryl transfer is known to occur[46,47]. But, we assumed that the hydrolysis of 1-pHis is faster than the transfer to another histidine and thus do not describe the transfer. We determined the half-lives of the phosphorylated species from the fitted rate constants that could not be determined empirically. In line with the literature[43,48], it became clear that any phosphate group at the 1-pHis

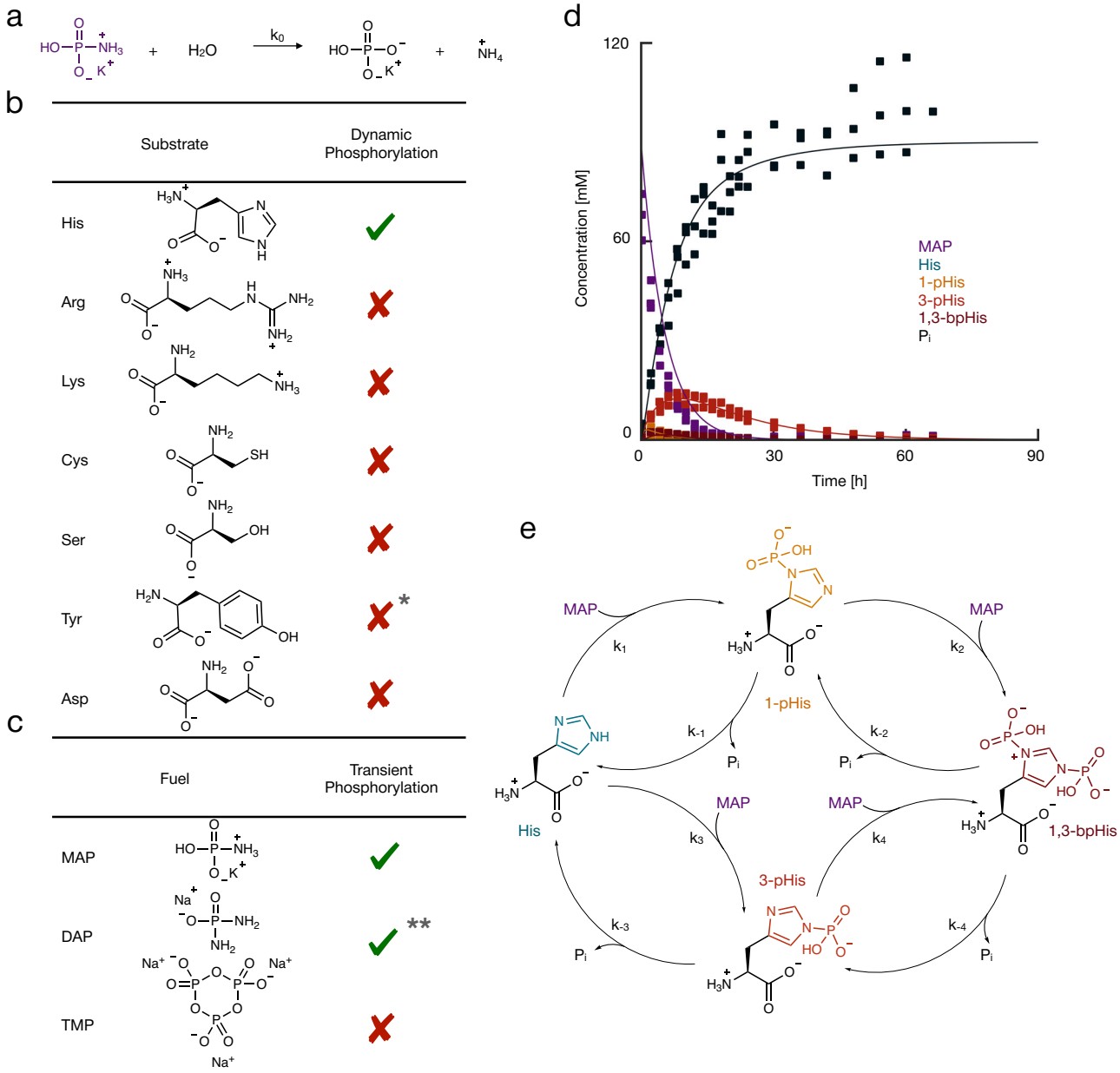

**Fig. 2 | Dynamic phosphorylation of amino acids using simple fuels. a** The hydrolysis of the phosphorylating agent MAP. **b** The amino acid-based catalysts were tested to convert the phosphorylating agent MAP. *pH 10.5 of Tyr containing sample due to low solubility at pH <pK$_a$. **c** The phosphorylating agents tested to phosphorylate histidine. **Not suitable due to many side reactions. **d** The concentration profiles as a function of time. The markers represent data determined by NMR, and the solid line represents a kinetic model. **e** The reaction network of the cycle with 1-, 3-pHis, and 1,3-bpHis as observed phosphorylated species. All experiments were performed in triplicate, and the conditions used were 75 mM amino acid, 80 mM of MAP, 50 mM DAP or 100 mM TMP in a 500 mM MES buffered solution at pH 6.5.

position was most labile. In contrast, 3-pHis was less labile, with an empirically determined half-life of $10.6 \pm 0.7$ h. The activation rate constants for the different nitrogen atoms in the histidine were in the same range. We confirmed the cyclic properties of this system with refueling experiments (Supplementary Fig. 5). Taken together, the MAP is catalytically converted by histidine, resulting in multiple transiently phosphorylated species. 3-pHis is the most long-lived, so its observed yield is the highest.

We embedded histidine in a peptide sequence and tested its reactivity to couple the reaction cycle to function in the future, for example, for producing active protocells (*vide infra*) or for forming non-equilibrium self-assemblies. The most straightforward peptide design was placing the histidine between two glycines (G) and

acetylating the NH$_2$-terminus (Ac-GHG-OH, Fig. 3c). We added MAP to 75 mM Ac-GHG-OH under the same conditions as above and monitored the concentration profiles with $^{31}$P-NMR over 90 h. Ac-G(3-pHis)G-OH appeared first, followed by Ac-G(1-pHis)G-OH, whereas in contrast to His, the bisphosphorylated species was not observed (Supplementary Fig. 7).

Because we did not observe the bisphosphorylated species, we neglected it in the kinetic model. The half-life of Ac-G(3-pHis)G-OH was experimentally determined to be $59.7 \pm 6.4$ h (Supplementary Table 5). Thus, for both histidine and Ac-GHG-OH, the 3-pHis isomer was the most stable. Yet, Ac-G(3-pHis)G-OH's deactivation was drastically slower than that of 3-phosphohistidine (Fig. 3a), which aligns with our expectations[49,50].

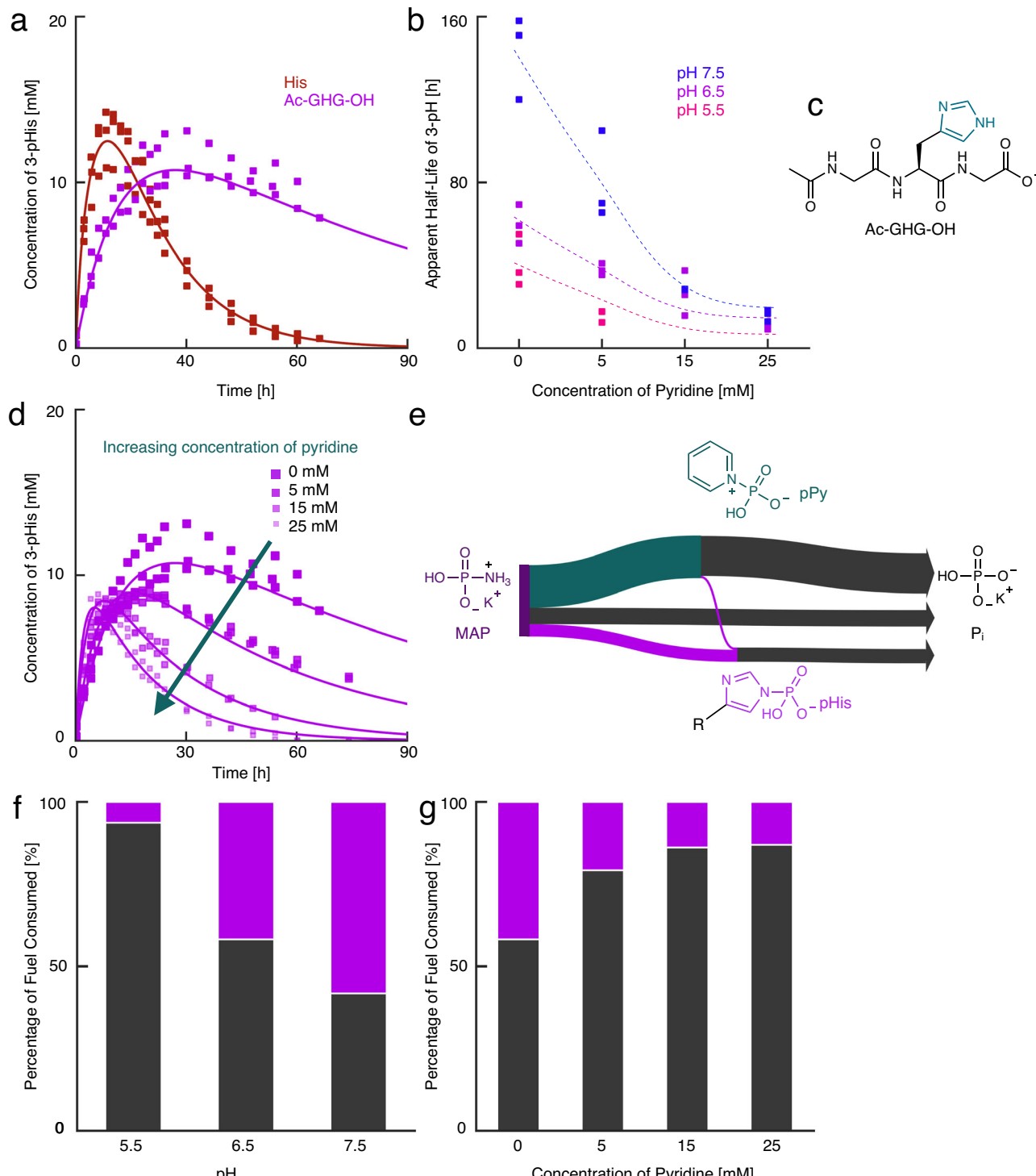

**Fig. 3 | Parameters affecting the dynamic phosphorylation of peptides. a** The evolution of the concentration of 3-pHis for Ac-GHG-OH (purple) and His (red). **b** Addition of pyridine and pH change the observed half-life of 3-pHis. It is increasing with increasing pH and decreasing amount of pyridine. **c** The structure of the peptide Ac-GHG-OH used in this study. **d** Representative concentration profiles of 3-pHis at pH 6.5 with increasing concentrations of pyridine. The smaller the squares, the shorter the observed half-life. **e** Sankey diagram to visualize the phosphate flow in the system (5 mM pyridine, pH 6.5). **f, g** The percentage of fuel converted to phosphate directly and through pyridine (black) or phosphorylated histidine species (magenta) depends on the pH and the amount of pyridine added. All experiments were performed in triplicate, and the conditions used were 75 mM precursor, 80 mM MAP and pyridine in a 500 mM MES or MOPS buffered solution at pH 5.5, 6.5 or 7.5.

The slow deactivation increased the overall reaction time to an impractical duration of hundreds of hours, thus, we tested whether the overall cycle could be accelerated by introducing catalysts for the dephosphorylation—we varied the pH and added pyridine to the reaction solutions[51]. Specifically, we used pH 5.5, 6.5, and 7.5 with and without additional pyridine (5, 15, 25 mM). We monitored the concentration profiles in response to fuel (Supplementary Figs. 6–8) and calculated the half-life via the slope of the apparent first-order decay (without pyridine $k_{-3}$ Supplementary Table 5, with pyridine $k'_{3\text{-pHis}}$, Supplementary Table 7). At pH 5.5 without pyridine, the half-life of Ac-

G(3-pHis)G-OH was $40.8 \pm 9.5$ h, which increased to $59.7 \pm 6.4$ h at pH 6.5 and to $143 \pm 15$ h at pH 7.5 (Fig. 3b). These findings align with the known acid-lability of the P − N bond[41,47,52]. The observed half-life and, thus, overall cycle lifetime were reduced when the pyridine concentration increased. For example, from $59.7 \pm 6.4$ h without pyridine at pH 6.5 to $9.56 \pm 0.4$ h with 25 mM pyridine (Fig. 3b, purple). However, at these higher pyridine concentrations, the observed yield was low (Fig. 3d). Furthermore, the yield was too low at low pH (5.5) and high pyridine concentration to determine the apparent Ac-G(3-pH)G-OH half-life.

We adjusted our kinetic model to predict the evolution of the reaction cycle with the peptide as the catalyst in the reaction cycle. For the kinetic model with pyridine present in the system, we neglected the intermolecular phosphate-group transfer between pyridine and pyridinio-$N$-phosphonate (pPy)[53]. We empirically determined the reaction rate constants of the direct fuel hydrolysis and the 3-phosphoisomer hydrolysis. The formation ($k_{pPy}$) and hydrolysis ($k_{-pPy}$) of pPy were determined by reacting the fuel with pyridine alone and fitting the data using COPASI (Supplementary Table 5). We fixed those rate constants in the kinetic model and fitted the remaining others (more information about the value and limits of the model can be found in the Supplementary Methods B).

The kinetic model allows us to predict through which pathways the fuel was converted into inorganic phosphate, which we demonstrate in a *Sankey* diagram (Fig. 3e). At pH 6.5 and with 5 mM pyridine, 17% of the MAP was used to phosphorylate histidine directly. Most of the MAP was used to phosphorylate pyridine, and only a small fraction was transferred to the histidine. The majority of the MAP is hydrolyzed directly or hydrolyzed by phosphorylating pyridine. We used these calculations to produce the plots in Fig. 3f, g, which show the pathway through which MAP is hydrolyzed. Either it is hydrolyzed directly or through phosphorylating pyridine (black and unwanted) or it is hydrolyzed through phosphorylating histidine (purple and wanted). At low pH values, the cycle was inefficient−only 1% of fuel was used to make the transient 1-pHis isomer and 5% to the 3-pHis isomer at pH 5.5 (Fig. 3f). The remainder of the fuel was lost through direct hydrolysis of MAP. The efficiency was much greater at a higher pH of 7.5, where 58% was used for the activation phosphorylation of histidine (1-pHis, 3-pHis, and 1,3-bpHis). Thus, decreasing the pH accelerates the total reaction cycle, but at the cost of efficiency. Adding pyridine has similar consequences for the reaction−the more pyridine added, the shorter the half-life of the phosphorylated histidine species, but at the cost of efficiency (Fig. 3g). For example, at pH 6.5 and without pyridine, 42% of our fuel was used to phosphorylate the histidine, whereas, with 25 mM pyridine, this number dropped to 13%.

We designed two histidine-bearing peptides to form active complex coacervate droplets upon dynamic phosphorylation (Supplementary Fig. 9). In this context, active means that the droplet material is controlled by two chemical reactions, *i.e.*, activation and deactivation (Fig. 4a)[54]. The successful peptide was peptide 1 (Ac-Tyr(OMe)-Asp-His-Asp-Asp-NH$_2$) as the polyanion, which we used with the peptide R$_{30}$ (NH$_2$-Arg$_{30}$-OH) as a polycation (Fig. 4b). At pH 7.5, peptide 1 carries three negative charges (−3). In this state, we expect the polyanion to have a low affinity for the polycation. We performed an isothermal calorimetry (ITC) study to corroborate this hypothesis. We found the peptide has a weak affinity for R$_{30}$ with a binding constant of $1.19 \pm 0.02$ mM (Supplementary Fig. 10). It can form complex coacervate-based droplets, but only at very high concentrations. Inspired by the design rules of polyanion and polycation-based complex coacervation systems described earlier[55], we anticipated that the dynamic phosphorylation of the histidine in peptide 1 would increase its affinity for the polycation by increasing its overall charge from −3 to −5[46].

We tested for the formation of active complex coacervate droplets by mixing 20 mM peptide 1, 12.5 mM MAP, 50 mM R$_{30}$ (charges), and 200 nm sulforhodamine B in 100 mM 3-($N$-morpholino)propane

sulfonic acid (MOPS) buffer at pH 7.5. We produced microreactors in which the droplets could be analyzed by mixing the aqueous solution with a perfluorinated oil. We used microreactors to analyze the droplets because, unlike glass slides (Supplementary Fig. 11), the droplets do not adhere to the microreactor walls. In addition, the evolution of the droplets' volume in a confined volume can be better tracked. We used a microfluidic droplet generator to produce microreactors of reaction solution surrounded by perfluorinated oil that we analyzed by confocal microscopy (Fig. 4c and Supplementary Fig. 12). As the concentration of phosphorylated peptide 1 increased, we witnessed the first coacervate-based droplets after 9 h. The droplets were round and rapidly fused, indicating their liquid nature (Supplementary Movie 1). Besides, a Fluorescence Recovery After Photobleaching (FRAP)-assay on Cy5-labeled R$_{30}$ further corroborated their liquid nature (Fig. 4d, e, Supplementary Fig. 13). The droplet rapidly recovered after bleaching with a half-life time ($t_{1/2}$) of $2.64 \pm 0.39$ s from which we could derive a diffusivity constant (D) of $0.295 \pm 0.04$ $\mu m^2$ $s^{-1}$.

After all droplet material had fused to one large droplet, it sunk to the bottom of the reactor. We tracked the average volume of the droplet material in every microreactor with time. From these measurements, we derived the droplet volume relative to its reactor. This fraction increased in the first 47 h to 1.5% as the reaction cycle produced increasing amounts of phosphorylated peptide 1 (Fig. 4f). Over the next hundred hours, it decayed until no droplets were found after 243 h.

We analyzed the kinetics of the reaction cycle by high-performance liquid chromatography (HPLC) and $^{31}$P-NMR and refitted our kinetic model to the obtained data (Fig. 4f, Supplementary Figs. 14 and 15). From the kinetic model, we can derive that the droplets dissolved when the total concentration of phosphorylated species fell below 262 $\mu M$. We can also conclude that the droplets, after roughly 30 h, can only comprise 3-pHis as a phosphorylated species because 1-pHis has already been hydrolyzed. Next, we tested how the chemical fuel can affect the macroscopic behavior of the active droplets using optical density at 500 nm as a readout. To produce sufficient droplet material for a reproducible increase in optical density, we added 2.5% poly(ethylene glycol) with an average molecular weight of 8000 g mol$^{-1}$ (PEG$_{8000}$) as a crowding agent[56]. We added various amounts of MAP as fuel. Above 10 mM of MAP, we found evidence of the first droplets. We then determined the time needed for the optical density to pass a threshold which we define as the lag time (Fig. 4g and Supplementary Fig. 16). With increasing amounts of fuel, the lag time decreased from $8.6 \pm 1.17$ h with 10 mM MAP to $2.55 \pm 0.13$ h with 50 mM of fuel. Notably, the decrease in the lag time tended to level off beyond 40 mM of fuel. With increasing fuel, the maximum turbidity reached, as a measure for the maximum amount of droplet material, increased too (Fig. 4h and Supplementary Fig. 16). To emphasize the cyclic properties of our system, we refueled the solution with 12.5 mM MAP after seven days. We observed an increase in the optical density. The turbidity reached a lower maximum intensity than the first cycle, presumably due to the accumulation of waste (Supplementary Figs. 16 and 17).

Our experiments show that the new reaction cycle we introduced can be used to create active droplets. In future works, we envision functionalizing the droplets with catalysts and reagents so that they can control their fate. For example, a catalyst could inhibit or accelerate dephosphorylation. To demonstrate that such an approach is viable, we tested whether other components, besides the two peptides, could partition into the droplets (Fig. 4i). We monitored the partitioning by confocal microscopy of Cy3-labeled single and double-stranded DNA, single-stranded RNA, and polystyrene sulfonate. From a qualitative point of view, both the single-stranded DNA and RNA partitioned well and homogeneously throughout the droplets. Excitingly, the double-stranded DNA and polystyrene sulfonate partitioned well and seemed to form microdomains of high concentration within the droplets.

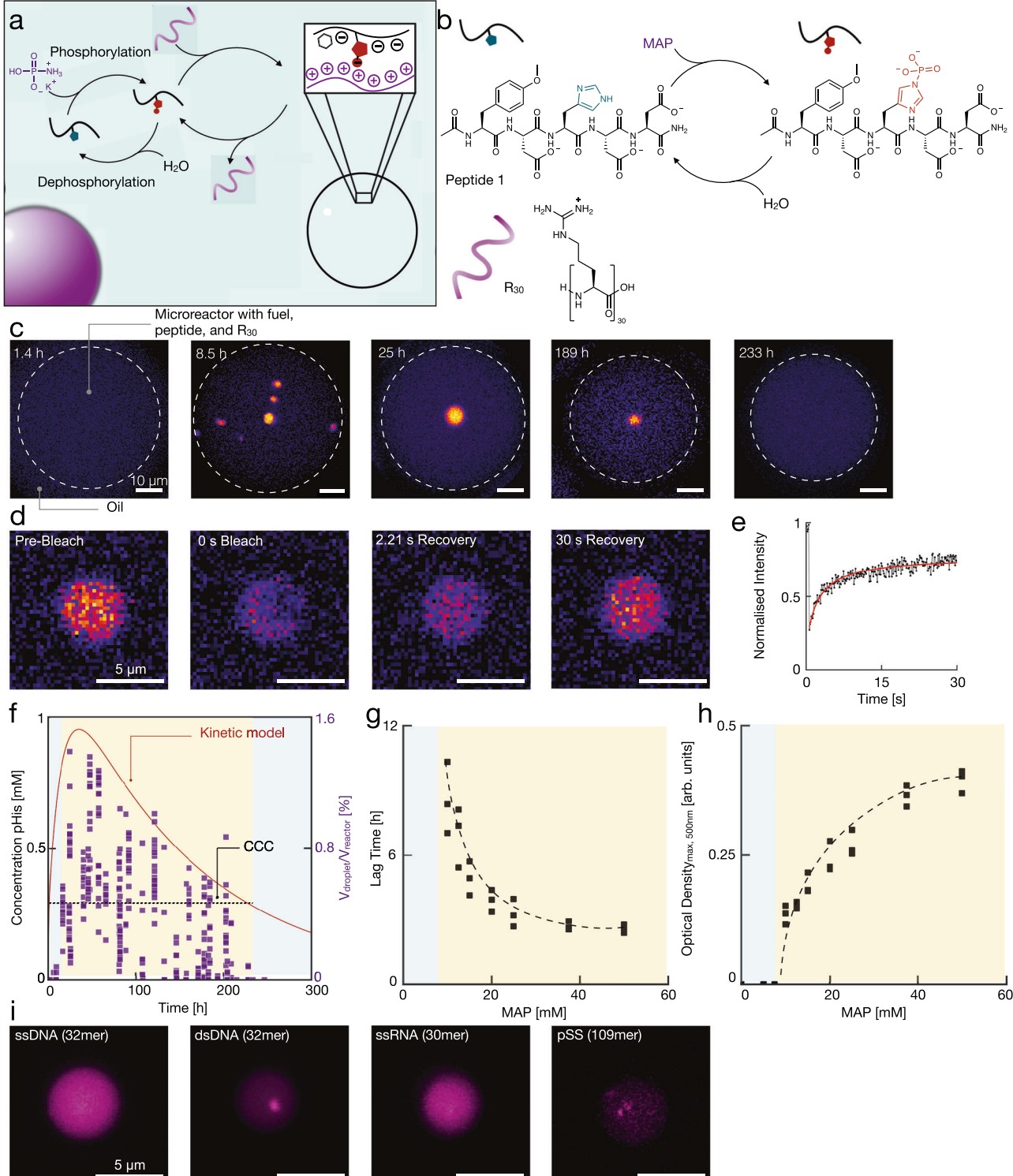

**Fig. 4 | Dynamic phosphorylation of a peptide forms active compartments. a** A scheme of the chemical reaction cycle combined with complex coacervation-based compartment formation with poly-arginine. **b** The dynamic phosphorylation-based chemical reaction cycle. **c** Representative micrographs at various times after adding MAP as fuel. The dotted line represents the water droplet. The micrographs are a maximum z-projection of a z-stack with a pseudocolor-coding. **d** Image sequence of a FRAP experiment. 500 nM Cy5-$R_{30}$ was used as a fluorescently labeled peptide. **e** Normalized intensity of the FRAP experiment over time. The fitting function is shown in red. **f** The sum of the concentration of 1- and 3-pHis as a function of time predicted by the kinetic model (left Y axis) overlaid on the relative total droplet volume against time measured from the confocal microscopy data. The experiments were carried out in triplicate. The critical coacervation concentration (0.26 mM) is highlighted with the dotted line. **g, h** Lag time and maximum intensity as a function of MAP concentration. 2.5% $PEG_{8000}$ was added to the reaction solutions as a crowding agent. All experiments were performed in triplicate, and all data is shown. **i** Micrographs of the active droplets in the presence of 200 nM polyanions of various structures (Cy3 labeled ssDNA, dsDNA, RNA, or pSS). Unless mentioned otherwise, the conditions used were 20 mM peptide 1, 12.5 mM MAP, 50 mM $R_{30}$ (charges) and 200 nM sulforhodamine B in a 100 mM MOPS buffered solution at pH 7.5.

Dynamic phosphorylation in biology is critically important in regulating the function of countless enzymes and metabolic pathways. Inspired by such biological dynamic phosphorylation, we introduced an enzyme-free reaction cycle that phosphorylates histidine as an amino acid or in a peptide. We coupled the reaction cycle to a function by designing a peptide to form active droplets upon phosphorylation. Analogous to how biomolecular condensates are regulated by posttranslational modification, we show that these droplets are dynamically controlled by the reaction cycle and can take functional molecules like DNA and RNA. Given the ubiquitous use of phosphorylation, we expect our reaction cycle to be powerful for regulating peptide self-assembly or dimerization, for example, in forming coiled coils or beta-sheets. Such phosphorylation-induced organization can be a powerful model for testing biological organization without the constraints of enzymes, or it can be used to drive molecular machinery, for example, in nanomotors and pumps that perform work by converting phosphorylating agents. Finally, we envision that the active droplets that exist at the expense of a prebiotically plausible fuel could be a protocell model or a platform for the synthesis of life. These protocells compete for fuel, which serves as a selection mechanism. In future work, such protocells would need a genotype that affects their phenotype, like self-replicating RNA, to have an advantage over their competition.

## Methods

### Materials
All chemicals were purchased from Sigma-Aldrich, VWR, or TCI Europe and used without further purification. General solvents were purchased from Sigma Aldrich in analytical or synthesis grade and used without further purification. Dimethylsulfoxide-$D_6$ in glass ampoules, Sylgard 184 silicon elastomer and curing agent were purchased from Sigma-Aldrich. 3 M™ Novec™ 7500 oil (dOil) and 2% fluorosurfactant (dSURF) were purchased from Fluigent. The peptide $NH_2$-$R_{30}$-OH ($R_{30}$) chloride salt was purchased from CASLO Aps.

### Synthesis of potassium monoamidophosphate (MAP)
The MAP synthesis was based on a protocol from Ying-Fei Wei and Harry R. Matthews[57]. Into 75.0 mL ice-cold 10% ammonium hydroxide solution 5.00 mL phosphorus(V)oxychloride (32.6 mmol) were added dropwise over 20 min and stirred for 15 min on ice. 250 mL acetone were added and then stirred for five more minutes at room temperature (RT). The mixture was transferred to a separation funnel, and after 5 min the phases were separated, and the bottom one was collected. A white precipitate crystallized while the collected water phase was acidified with 2.00 mL acetic acid to pH 6. The mixture was stored in the fridge (4 °C) for 20 min to facilitate crystallization and 20 min at RT. The ammonium monoamidophosphate was filtered off and washed with ice-cold 10.0 mL absolute ethanol and 5.00 mL diethyl ether. It was dried overnight in the desiccator, and 1.00 g $NH_4PO_3NH_2$ was obtained (8.77 mmol, 27%). The ammonium monoamidophosphate was dissolved in 2.00 mL 50% KOH (w/v) solution and heated for 10 min at 60 °C. The solution was cooled to 5–10 °C in a water bath and acidified with 0.75 mL acetic acid to pH 6 with acetic acid. The mixture was added to 100 mL of absolute ethanol and crystallized for 1.5 h at RT. Potassium monoamidophosphate was filtered off. It was later washed with ice-cold 5.00 mL absolute ethanol and 4.00 mL diethyl ether and dried in the desiccator overnight. 371 mg of MAP were obtained (2.75 mmol, yield = 31%). It was characterized by $^{31}$P-NMR and elemental analysis (Supplementary Methods A).

### Synthesis of diamidophosphate (DAP)
The DAP synthesis was based on a protocol from Krishnamurthy[58]. 5.00 g phenyl phosphorodiamidate (29.1 mmol) were added to 15.0 mL 4 M sodium hydroxide solution and stirred for 10 min at 110 °C. The

solution was concentrated and 70 mL absolute ethanol were added at 0 °C. The precipitate was dissolved in 50 mL water after filtration and washed with dichloromethane and ethyl acetate. The concentrated aqueous solution was added dropwise to 50 mL ice-cold absolute ethanol under stirring. A white precipitate was formed and filtered off. It was later washed with 10 mL ice-cold absolute ethanol 3 times and dried in the desiccator overnight. 1.21 g of DAP were obtained (10.3 mmol, yield = 35%). It was characterized by $^{31}$P-NMR and elemental analysis (Supplementary Methods A).

### Manual solid-phase peptide synthesis under controlled heating
All peptides were synthesized on a 0.5 mmol scale. Glycine preloaded Wang resin (0.5 mmol, 0.69 mmol g$^{-1}$ loading) was used to synthesize Ac-GHG-OH, aspartic acid preloaded Wang resin (0.5 mmol, 0.67 mmol g$^{-1}$ loading) for Ac-Y(OMe)DHDD-OH and rink amide resin (0.5 mmol, 0.29 mmol g$^{-1}$ loading) for Ac-Y(OMe)DHDD-NH$_2$. The reaction vessel was connected to a nitrogen line, and the waste flask was connected to a water pump. The resin was swelled in dimethylformamide (DMF) with an $N_2$ stream for 20 min at room temperature. The peptide synthesis was performed at 68 °C.

Before each coupling step, the N-terminal fluorenylmethyloxycarbonyl (Fmoc)-protecting group was cleaved using 10–30 mL of a 5% (w/v) solution of piperazine in DMF (for D-containing peptides, an additional 0.2 M hydroxybenzotriazole (HOBt) was added). The reaction mixture was stirred with an $N_2$ stream for 1 min and 5 min. The deprotecting solution was removed, and the resin was washed with DMF. For each coupling, 3 eq. of amino acid were used together with 2.8 eq. of O-(1 H–6-Chlorobenzotriazole-1-yl)–1,1,3,3-tetramethyluronium hexafluorophosphate (HCTU) and 6 eq. of N,N-Diisopropylethylamine (DIPEA). HCTU and DIPEA were added to the amino acid and vortexed until dissolved. The coupling solution was added to the resin and stirred with an $N_2$ stream for 6 min. After the coupling, the solution was drained, and the resin was washed with DMF. The deprotection, washing, coupling, and washing cycle was carried out for each amino acid. After the last coupling, the peptide was acetyl end-capped at room temperature. Therefore, a final deprotection was conducted; the resin was washed at room temperature, and 6 eq. of acetic anhydride and 6 eq. of DIPEA in DMF solution were added to the resin and stirred with $N_2$ stream for 10 min at RT. The resin was washed with DMF and DCM. A cleavage solution consisting of 2.5% MQ-water, 2.5% triisopropyl silane (TIPS), and 95% trifluoroacetic acid (TFA) was prepared to cleave the peptide from the resin. It was added to the resin and agitated for two hours at RT. The cleavage solution was collected by filtration, and the resin was washed with dichloromethane (DCM). The solvents were removed by co-distillation under reduced pressure using a rotary evaporator (Hei-VAP Core, VWR). The crude peptides were dissolved in H$_2$O:ACN (70:30) and purified using a reversed-phase preparative HPLC (Thermo Fisher Dionex Ultimate 3000, Hypersil Gold 250 × 4.8 mm, Thermo Scientific™ Dionex™ Chromelon™ Chromatography Data System for Ac-GHG-OH; RP-HPLC on an Agilent 1260 Infinity ll setup, Agilent InfinityLab ZORBAX SB-C$_{18}$ column 250 mm×21.2 mm, 5 µm particle size for Ac-Y(OMe)DHDD-OH/-NH$_2$) in a linear gradient of H$_2$O:ACN each with 0.1% TFA (Supplementary Table 1, 2).

All peptides were lyophilized (Christ Freeze Dryer Alpha 2-4 LDplus, VWR) and stored at −20 °C. They were characterized by electrospray ionization mass spectrometry (ESI-MS, Thermo Scientific, LCQ Fleet ION Trap Mass Spectrometer) in positive or in negative mode, analytical HPLC (Vanquish DUO HPLC system (Chromeleon software version 7.2.10 ES) with a Hypersil-Gold, reversed-phase C$_{18}$ column (particle size: 3 µm, length: 100 mm, ID: 2.1 mm), eluted with a gradient of 0.1% TFA H$_2$O:ACN for Ac-GHG-OH; Vanquish SINGLE HPLC system (Chromeleon 7 Data System Software Version 7.3.1) with EC 150/4 NUCLEODUR C$_{18}$ Pyrmid, 3 µm (particle size: 3 µm, length: 150 mm, ID: 4 mm), Machery-Nagel; eluted with a gradient

triethylammonium acetate (TEAA) 25 mM and ACN:TEAA 25 mM 25:75 (pH 7.0) at 30 °C) detected at 220 nm and $^1$H NMR (Supplementary Tables 3, 4, Supplementary Methods A).

### Fluorescent labeling of R$_{30}$ with cyanine 5 (Cy5)

For the labeling of R$_{30}$ with Cy5, the protocol "Amersham CyDye mono-reactive NHS Esters - Reagents for the labeling of biological compounds with Cy monofunctional dyes - Product Booklet" published by cytiva was adapted. A solution of 1 mg Cy5-NHS ester in 50 μL DMF was prepared, adding 1 mL of a 50 mM NaHCO$_3$ buffered 62 μM R$_{30}$ solution. The reaction mixture was stirred for 24 h at room temperature. To purify the labeled peptide, it was dialyzed (Thermo scientific Slide-A-Lyzer Dialysis Cassette, 2000MWCO, 2.0 −12 mM capacity) against 2.5 mL water (12 h and 24 h) and lyophilized afterward. A degree of labeling of 0.2609 by applying *Lambert-Beer*'s law with an absorbance at 650 nm was obtained. The extinction coefficient of Cy5 is 250000 M$^{-1}$ cm$^{-1}$.

### General sample preparation

2-(*N*-morpholino)ethane sulfonic acid (MES) and 3-(*N*-morpholino) propane sulfonic acid (MOPS) buffers were prepared by dissolving MES hydrate or MOPS in MQ-water (11% D$_2$O) to give a 500 mM solution. 5 M NaOH was added to adjust the final pH of 5.5, 6.5 for MES, and 7.5 for MOPS.

### Sample preparation for kinetic analysis

Stock solutions of the precursor (75 mM His, Arg, Lys, Ser, Cys, Asp, Ac-GHG-OH) were prepared in 500 mM MES or MOPS buffer and adjusted to respective pH using NaOH or HCl, whereas 200 mM stock solutions of the coacervation peptides were prepared in water with 500 mM NaOH added. The 80 mM Tyr stock solution was prepared in 500 mM sodium carbonate buffer at pH = 10.5.

The 2 M stock solution of pyridine, the stock solutions of the polycation (500 mM R$_{30}$), and the 10% PEG$_{8000}$ stock solution were prepared in MQ-water and stored at 8 °C. Reaction cycles were started by the addition of the high concentration (2 M or 250 mM) MAP to the solution. All experiments were carried out at room temperature or 25 ( ± 0.5) °C.

### Kinetic model

A kinetic model written with COPASI 4.36 (Build 260) was used to predict the evolution of the reaction network over time (Supplementary Methods B). The constants were fitted to the $^{31}$P-NMR kinetic measurements, and the concentrations of compounds were calculated every five minutes. The rate constants are given in Supplementary Table 5.

### Elemental analysis

The CHNS values are determined simultaneously by combustion analysis in a EuroEA Elemental Analyser from HEKAtech. The following standard substances were included in the analyses: 2x BBOT and 1x chloro-2,4-dinitrobenzene. An error tolerance range of ± 0.3% can be specified. Potassium was determined using acid digestion and subsequent measurement on an Agilent 280 FS-AA atomic absorption spectrometer (flame AAS). Potassium dihydrogen phosphate was used as the reference substance. The error tolerance range is ± 0.5%. Phosphorus was determined after acid digestion and subsequent measurement on a Cary100 UV/VIS photometer from Agilent. Triphenylphosphine was used as the test substance. The error tolerance range is ± 0.3%. As obtained from the central analytics of the Technical University of Munich.

### $^1$H Nuclear magnetic resonance spectroscopy (NMR)

$^1$H-NMR spectra were recorded on a Bruker AV-500HD NMR spectrometer. All chemical shifts δ are given in parts per million (ppm) and referenced to the residual proton signal of dimethylsulfoxide-d$_6$ (δ = 2.50 ppm). The NMR spectra were analyzed using MestReNova© software (version 14.2.3-29241).

### $^{31}$P-NMR spectroscopy kinetic measurements

Over time, the fuel consumption, formation, and hydrolysis of phosphorylated compounds were monitored with $^{31}$P-NMR on a Bruker AV500CR NMR-spectrometer. An inverse gated H-decoupled[59] method with 16 scans and 25 s relaxation delay was used. As internal standard, phosphonoacetic acid (20 mM, δ = 15.7 ppm[60]) was used. The concentrations were calculated by using Eq. (1)[59].

$$c_{\mathrm{x}} = c_{\mathrm{standard}} \frac{\mathrm{integral}_{\mathrm{x}}}{\mathrm{integral}_{\mathrm{standard}}} \qquad (1)$$

With $c_{\mathrm{x}}$: concentration of compound x, $c_{\mathrm{standard}}$: concentration of the internal standard, integral$_{\mathrm{x}}$: integral of compound x, integral$_{\mathrm{standard}}$: integral of internal standard.

For $^{31}$P-NMR kinetic measurements, 200 μL (Ac-GHG-OH) or 500 μL (amino acids, hydrolysis) of the samples were prepared. The samples consisted of 500 mM MES or MOPS buffered solution of 75 mM amino acid (sodium carbonate for 60 mM Tyr) or Ac-GHG-OH in the presence and absence of pyridine, and the addition of 80 mM MAP started the reaction. The hydrolysis of MAP at different pH and fuel consumption, respectively, and the formation of the transient species were determined by applying Eq. (1). The NMR spectra were analyzed using MestReNova© software (version 14.2.3-29241).

To determine the reaction rate constants k$_{\mathrm{py}}$ at different pH values, samples consisted of 500 mM MES or MOPS buffered solution of 80 mM MAP in the presence of 15 mM pyridine (pH 5.5 and 6.5) and 5 mM (pH 7.5).

### pH measurements

A pH meter (HANNA HI 2211 pH/ORP Meter) with an Ag/AgCl electrode was used for the pH adjustments of the stock solutions and pH value measurements. It was calibrated with standard calibration solutions (pH = 7.01 and 4.01) each time before usage.

### Electrospray ionization-mass spectrometry (ESI-MS)

An LCQ Fleet Ion Trap Mass Spectrometer (Thermo Scientific) was used for ESI-MS experiments. The data was evaluated using the Thermo Xcalibur Qual Browser 2.2 SP1.48 software. The fractions of the preparative HPLC were collected, and 2 μL was injected directly into the loop.

### Isothermal titration calorimetry (ITC) measurements

ITC experiments were performed with a MicroCal PEAQ-ITC from Malvern Pananalytical. All experiments were performed at 25 °C. The following conditions were used: R$_{30}$ (15 mM charges, 500 μM strand in MOPS 100 mM, pH 7.5) was titrated with the Ac-Y(OMe)DHDD-OH (10 mM in 100 mM MOPS, pH 7.5): 26 injections, 3 μL each. A control was performed by titrating the corresponding amount of peptide in 100 mM MOPS buffer (pH 7.5) and used to correct for dilution enthalpy. Data were fitted to a single set of sites binding isotherm using the PEAQ-ITC Analysis software.

### Analytical high-performance liquid chromatography (HPLC) kinetic measurements

For the kinetic measurement of the transient 1- and 3-phospho isomers of the Ac-Y(OMe)DHDD-NH$_2$, an analytical HPLC (Vanquish SINGLE HPLC system with EC 150/4 NUCLEODUR C$_{18}$ Pyrmid, 3 μm (particle size: 3 μm, length: 150 mm, ID: 4 mm), Machery-Nagel; eluted with a gradient of TEAA 25 mM and ACN:TEAA 25 mM 25:75 (pH 7.0) at 30 °C) was used. The Chromeleon 7 Data System Software (Version 7.3.1) was used to evaluate the received data.

For the experiments, samples consisted of 20 mM Ac-Y(OMe) DHDD-NH$_2$ and 12.5/25 mM MAP in a 100 mM MOPS buffered solution. The reaction was started by adding MAP, and the reaction solution was transferred to inserts for a 1.5 mL screw cap HPLC glass vial. 1 μL of the solution was injected without any further dilution. A linear gradient of TEAA 25 mM and ACN:TEAA 25 mM 25:75 (pH 7.0) at 30 °C column temperature was used.

Calibration curves for the peptide (in MQ water) were performed in triplicate with the corresponding method (Supplementary Methods A and Supplementary Table 3, 4).

## UV/Vis measurements

The UV/Vis measurements were carried out using a Multiskan FC (ThermoFisher) microplate reader. Samples (50 μL) were prepared in Eppendorf tubes and transferred into a 96 half-area well-plate (tissue culture plate, non-treated). The temperature (25 ± 0.5 °C) was set 10 min before starting the measurement. Each experiment was performed at 500 nm, and every 5 min, a data point was acquired. (n = 3)

For absorbance/optical density experiments, 50 μL sample volumes were prepared. The samples consisted of 20 mM Ac-Y(OMe) DHDD-NH$_2$, 50 mM R$_{30}$ (in positive charges), 0, 5, 7.5, 10, 12.5, 15, 20, 25, 37.5, 50 mM MAP (+x mM P$_i$) in a 100(-x) mM MOPS pH 7.5 buffered solution. To sustain the ionic strength of the system for samples fueled with less than 12.5 mM MAP, the missing amount of P$_i$ was added, and for samples fueled with more than 12.5 mM MAP, the buffer concentration was reduced (buffer and Na+ were considered).

## Confocal fluorescence microscopy

A Leica SP5 confocal microscope using a 63x oil immersion objective was used to image the droplets in microreactors. Coacervates were dyed with the fluorophore Sulforhodamine B and Cy5- R$_{30}$, which were excited at 561 nm and 638 nm and detected from 566-642 nm and 643–839 nm with a HyD detector. The pinhole was set to 1 Airy unit. Micrographs of the microfluidic droplets were acquired in z-stacks with 1 μm between z-planes to analyze the evolution of coacervates in the entire microfluidic droplets. The micrographs were recorded with a resolution of 1024 ×1024 pixels or 512 ×512 pixels at 1x zoom and 600x scan speed (bidirectional scan). Measurements were performed at 21 °C, but the samples were incubated at 25 °C. ImageJ (Version 2.14.0) was used to analyze the micrographs.

For confocal experiments, 25 μL (snipping method) or 150 μL (microfluidic setup) sample volume was prepared. If not mentioned differently, the samples consisted of 20 mM Ac-Y(OMe)DHDD-NH$_2$ (15 mM Ac-Y(OMe)DHDD-OH), 50 mM R$_{30}$ (positive charges), 200 nM sulforhodamine B, 500 nM Cy5-R$_{30}$ and 12.5 mM MAP/P$_i$ in a 100 mM MOPS buffered solution. First, MOPS, peptide, and dyes were combined, and the reaction started with adding MAP. After the agitation of the reaction, solution R$_{30}$ was added, and it was proceeded with the water in oil emulsion preparation.

For the partition experiments of the non-functional structures instead of dyes, 200 nM of the Cy3-labeled structures were added and the samples were prepared with the snipping method (Supplementary Table 8).

## Microfluidic chip preparation

Microfluidic Polydimethylsiloxane (PDMS)-based devices were designed with QCAD-pro (RibbonSoft GmbH) and fabricated using photo- and soft-lithography[61,62]. For the molding of the PDMS, a 10:1 (w/w) mixture of PDMS prepolymer (Sylgard™ 184 Silicone Elastomer Base, Dow Corning) and curing agent (Sylgard™ 184 Silicone Elastomer Curing Agent, Dow Corning) was stirred vigorously and degassed under vacuum for 30 min. The mixture was poured into the master, consisting of a silanized silicon wafer in a Petri dish, and cured for 4 h at 60 °C. After that, the PDMS replica was cut with a scalpel to fit on 60 mm × 24 mm coverslips and peeled from the master. The in- and

outlets were punched into the PDMS with punchers (WellTech Rapid-Core Microfluidic Punches 0.5 mm). The PDMS chip and coverslip were cleaned in a 33% (v/v) neutral Extran® MA 02 solution in an ultrasonic bath for 15 min and washed twice with MQ water in an ultrasonic bath for 15 min. The PDMS chip and the coverslip were dried with a nitrogen flow, placed in a plasma cleaner (Harrick Plasma, Plasma Cleaner Model PDC-002-CE), and oxidized for 2 min. After that, the PDMS chip and the coverslip were immediately brought into contact under light pressure, cured at 60 °C overnight, and sealed irreversibly. The channels were coated with Sigmacote® for 5 min and subsequently rinsed with water.

## Water in oil emulsion preparation

The water in oil emulsions was prepared either with a microfluidic droplet generator or with a snipping technique. HFE oil with 1.33% fluoro-surfactant was used for both types of sample preparation. A pump setup for microfluidics and a microfluidic chip were used to obtain standardized water droplet sizes. The chip has two inlets for the oil and the reaction mixture. A pressure of 610 mbar was applied to both channels, and the emulsion was collected from the outlet. For the snipping method, 5 μL of the reaction solution was added to 50 μL of the HFE oil with 1.33% fluoro-surfactant and snipped against the tube. Both emulsions were pipetted below a glass slide, glued onto another glass slide with double-sided sticky tape, and sealed with two-component glue.

## Fluorescence recovery after photobleaching (FRAP)

A Leica SP8 confocal microscope with a 63x water immersion objective was used to perform FRAP experiments. The samples were prepared as described previously described. Cy5- R$_{30}$ was bleached with a 638 nm laser and imaged at 648–784 nm with a PMT. The pinhole was set to 1 Airy unit, and the images were acquired at a resolution of 254 ×254 pixels. Recovery data was background and photofading corrected by Eq. (2)[63]

$$I_{corrected}(t) = \frac{I_{raw}(t) - I_{background}(t)}{I_{fading}(t) - I_{background}(t)} \qquad (2)$$

With $I_{corrected}(t)$: background and photofading corrected fluorescence intensity at time point t, $I_{raw}(t)$: raw fluorescence intensity at time point t, $I_{background}$: fluorescence intensity of background at time t, and $I_{fading}(t)$: fluorescence intensity of a neighboring not bleached droplet at time point t.

The corrected intensities were normalized by the prebleach intensity using Eq. (3)[63]

$$I_{normalized}(t) = \frac{I_{corrected}(t)}{I_{prebleach}} \qquad (3)$$

With $I_{normalized}(t)$: normalized fluorescence intensity, $I_{prebleach}$: fluorescence intensity before bleaching.

For the fit of the FRAP recovery time trace Eq. (4) was used[64]

$$I_{normalized}(t) = \frac{a + b(\frac{t}{\tau_{1/2}})}{1 + \frac{t}{\tau_{1/2}}} \qquad (4)$$

With a, b, and $\tau_{1/2}$: fitting parameters. $\tau_{1/2}$: recovery half-time.
The diffusivity constant was calculated using Eq. (5)[64].

$$D = \frac{r^2}{\tau_{1/2}} \qquad (5)$$

With $D$: diffusivity constant and r: radius of the bleach spot (r was determined by the region of interest used as $I_{raw}(t)$).

## Data availability

The data generated in this study are provided in the Source Data file. All other data are available from the corresponding author upon request. Source data are provided with this paper.

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

## Acknowledgements

The BoekhovenLab is grateful for support from the TUM Innovation Network - RISE funded through the Excellence Strategy and the European Research Council (ERC starting grant 852187). This research was conducted within the Max Planck School Matter to Life, supported by the German Federal Ministry of Education and Research (BMBF) in collaboration with the Max Planck Society. Funded by the Deutsche Forschungsgemeinschaft (DFG, German Research Foundation) under Germany's Excellence Strategy - EXC-2094 – 390783311. We are grateful for the help and support of Dr. Alexander Bergmann with the microfluidic setup, Apl. Prof. Dr. Wolfgang Eisenreich and Dr. Thomas Geisberger with the $^{31}$P-NMR, Anna-Lena Holtmannspötter with the partitioning, and Marvin Sam Batshoun with data analysis.

## Author contributions

S.M.P. and J.B. conceived the presented idea. S.M.P., M.S., and J.B. designed the experiments. S.M.P., M.S., B.A.K.K., M.W., and J.S. carried out the experiments. J.B. and S.M.P. wrote the manuscript. J.B. supervised the work.

## Funding

## Competing interests

The authors declare no competing interests.

## Additional information

**Peer review information** : *Nature Communications* thanks Scott Hartley and the other, anonymous, reviewer(s) for their contribution to the peer review of this work. A peer review file is available.

