## [Peer Review File · Nature Communications]

Active droplets through enzyme-free, dynamic phosphorylationREVIEWER COMMENTS

Reviewer #1 (Remarks to the Author):

Key results: The authors describe a strategy to transiently phosphorylate histidine, with the goal of mimicking the transient activation of enzymes in cells. Monoamidophosphate (MAP) is the main phosphorylating agent, and water is the dephosphorylating agent. Histidine catalyses the hydrolysis of MAP, getting phosphorylated/de-phosphorylated in the process. When histidine is embedded in a tripeptide, de-phosphorylation proves to be too slow for the reaction cycle to be considered dynamic, and the authors deploy pyridine to accelerate it. In the presence of MAP and pyridine, the half-life of the phosphorylated peptide is reduced to a few hours. Experimental data and kinetic modeling are used to depict the flow of phosphate in the system. The authors show that when the reaction is carried out in the presence of a polycation, supramolecular assemblies (coacervates) form and disappear following the dynamic phosphorylation of histidine.

Validity & Significance: The experimental data in the manuscript is complete and the kinetic model provides a good fit. However, I believe the concept and strategy described in this manuscript are not significantly novel, and would not have a sufficient impact in the community to grant publication in Nature Communications. Transient compartmentalization controlled by phosphorylation has been shown previously (DOI: 10.1038/nchem.2414, 10.1039/c7sm01897e). It is true that the manuscript provides a non-enzymatic version, but still the applicability is limited by the fact that the reaction only works with the pair MAP/histidine (tables in Figure 1). Recent literature on prebiotic phosphorylation tends to look for agents with a broader spectrum (e.g. DOI 10.1021/jacs.3c08539, 10.1038/s41467-021-25555-x); or in the case of histidine itself, for functional reactions with greater resemblance to natural catalytic cycles (see alternative with imidazole phosphate in Maguire ChemRxiv 2023). MAP, as the authors point out themselves, is not the most prebiotically plausible phosphorylating agent; and the use of pyridine as a substrate (that is mostly phosphorylated directly and competes with histidine) is somewhat confusing and undermines, in my opinion, its potential impact in the prebiotic chemistry field. The results on the formation of compartments are also weakened by the use of PEG 8k (at 2.5% concentration), a crowding agent known to promote and interfere with phase separation –

more than a simple turbidity enhancer. Combined with the fact that most micrographs show only one droplet, or one microreactor, it indicates that the supramolecular results may be not robust or reproducible, and that the impact on the protocell/dynamic assemblies community would be very limited.

I include below some specific minor and major concerns about the manuscript.

Clarity and context:

1. The authors use terms such as "active droplets" in the title, "self-sustaining complex coacervate droplets", but do not justify it in the text or in the results. Active droplets would imply a non-equilibrium behavior, which is not clear to me in the data (the droplets emerge, coalesce, and dissolve). Self-sustaining would imply that the droplets facilitate the phosphorylation reaction that leads to their own formation, which was not probed in the manuscript.
2. The description of the reaction as a cycle, or as a network (lines 64, 129), may be misleading, with the deactivation agent being the reaction's solvent (or pyridine), and the fuel MAP not being regenerated.
3. I am missing the explanation as to how short of a half-life is desirable in a biological or chemical context, and why the choice to rely on water or pyridine as de-activating agents, as opposed of some prebiotically relevant substrates. The authors could include a more specific discussion on histidyl kinases, instead of the general introduction on ATP.
4. Figures: scheme 1 would be more appropriate as a graphical abstract, it is taking up quite a lot of space but not adding as much information. Figure 1 could be reorganized to take up less space (the table format may not be necessary). Figure 2 contains some redundant data, panels F and G could be in supplementary information. Instead, a table for half-life and conversion% would be helpful. In Figure 3, what does Y'Dam mean?
5. The clarity of the manuscript could be increased by changing the acronyms. Instead of 1-pH, His1-P for example, as pH already has its own meaning.
6. Line 27: The sentence starting with "Does not occur..." is incomplete. The text from line 27 up to 39 needs to be restructured, it is confusing and not conveying its message – which I think is that biological cycles benefit from using ATP as a fuel due to it being thermodynamically unstable but kinetically stable.

7. Line 187: there is a typo, in "Either it is hydrolyzed directly or through phosphorylating histidine (unwanted)" it should be pyridine, not histidine. In line 192 there is a repetition "activation phosphorylation". This paragraph could be made clearer and the quantitative data mentioned could be included in a table elsewhere for clarity.

Data and methodology:

8. Both experimental and modeling kinetic data are well presented, error bars are shown but it is not mentioned if the mean is taken over multiple experiments or different measurements or time points. I appreciate how the methodology was extensively described.

9. Overall, some experiments could be better motivated in the text: for example, it is not clear to me that such a complete kinetic modeling is necessary, or how it can contribute to the understanding of the system. Another example is why the authors choose to conduct microscopy in microfluidically-produced micro-reactors, and not on a typical microscopy slide/chamber - what advantages to the paper's goals does it bring? The ITC and DNA partitioning data is included but not well integrated within the text - how is this data relevant to the manuscript?

10. The microscopy images do not immediately point to coacervate droplets (if you compare to other peptide/peptide complex coacervates in related literature), although I understand that FRAP supports it. Perhaps resolution and visibility would be increased without the micro-reactors. And instead of adding PEG, I would consider reducing buffer concentration.

11. Scale bars are often missing.

Reviewer #2 (Remarks to the Author):

Referee report

Active droplets through enzyme-free, dynamic phosphorylation

Authors: Simone Poprawa,¹ Michele Stasi,¹ Monika Wenisch,¹ Brigitte A. K. Kriebisch,¹ Judit Sastre,¹ Job Boekhoven¹

Reviewer: The article presents an elegant system for the construction of active droplets using dynamic phosphorylation in an enzyme-free manner, and hence furnishes a platform that others can readily built upon. This work will likely be picked up rapidly by several communities (for instance systems chemistry, active matter, and origins of life) and can be

expected to become a highly cited paper. As such, I am overall favorable to its publication in Nature Communications. Nevertheless, in terms of writing, presentation and contextualization, the present version feels unfinished and the scholarly quality of the work can be favorably improved.

In chronological order:

Abstract

Reviewer: The abstract provides hardly more information than the title about the content of the paper and remains vague in instances where being precise would hardly consume more space, for instance it may be pointed out that the system is specific to histidine, and what phosphorylating agent is used.

Authors: Enzyme-free, synthetic systems that use dynamic phosphorylation to regulate supramolecular processes do not exist

Reviewer: Do not exist is a confusing phrasing. The authors presumably mean that such systems 'have not yet been reported, to the best of our knowledge'.

Authors: Our new reaction cycle will be valuable as a model for biological phosphorylation but can also offer insights into protocell formation.

Reviewer: Given the long history of MAP and DAP chemistry and renewed recent interest (which should be cited), the possessive phrasing 'our new reaction cycle' feels unnecessary here. The article does not clarify in what sense(s) the reaction cycle may be expected to be deliver on these promises. Since the phosphorylation is specific to histidine, it is not a priori obvious that it provides a model for biological phosphorylation.

Introduction

Reviewer: The introduction explains at length certain aspects that are widely known, while assuming intimate familiarity of the reader with communities and highly specific branches of literature that are not cited. In particular, pertinent recent literature involving phosphates in systems chemistry and MAP/DAP in prebiotic chemistry appears to be overlooked.

With respect to the former, notably:

Spontaneous and Selective Peptide Elongation in Water Driven by Aminoacyl Phosphate Esters and Phase Changes, Kun Dai, Mahesh D. Pol, Lenard Saile, Arti Sharma, Bin Liu, Ralf Thomann, Johanna L. Trefs, Danye Qiu, Sandra Moser, Stefan Wiesler, Bizan N. Balzer, Thorsten Hugel, Henning J. Jessen, and Charalampos G. Pappas*, J. Am. Chem. Soc. 2023, 145, 48, 26086–26094, <https://doi.org/10.1021/jacs.3c07918>

A Ribonucleotide <-> Phosphoramidate Reaction Network Optimized by Computer-Aided Design. Andreas Englert, Julian F. Vogel, Tim Bergner, Jessica Loske, and Max von Delius. J. Am. Chem. Soc. 2022, 144, 33, 15266–15274

Authors: Does not occur on a reasonable time scale;

Reviewer: This phrase is grammatically incorrect.

Authors: Their need for highly evolved enzymes also offers limited insight into phosphorylation before life existed

Reviewer: This statement is suggestive and lacks nuance. It suggests an implicit historical claim that phosphorylation is a process that predates life. This is a popular – though unproven - conjecture in a niche of origins of life, where it is more broadly believed that modern biomolecules coincide with those present at abiogenesis. Since much of the prospective readership is not directly from the origins community and may not be aware of niche conjectures, any reference to them needs to be contextualized. If the authors choose to do so, they may also want to cite appropriate literature that lays out these conjectures and viewpoints thereon. A recent work on conjectured roles of phosphate groups in prebiotic chemistry is for instance: Ring Opening of Glycerol Cyclic Phosphates Leads to a Diverse Array of Potentially Prebiotic Phospholipids, Maiia Aleksandrova, Fidan Rahmatova, David A. Russell, and Claudia Bonfio. J. Am. Chem. Soc. 2023, 145, 47, 25614–25620, <https://doi.org/10.1021/jacs.3c07319>

It should be stressed that biochemistry-first as a conjecture has been questioned many times over, even by its proponents within the origins literature, a relatively recent source being:

Life's Biological Chemistry: A Destiny or Destination Starting from Prebiotic Chemistry? R. Krishnamurthy, Chem. Eur. J. 2018, 24, 16708.

A more nuanced phrasing would be ‘.. into the roles phosphorylation may potentially have

played before life existed’.

Authors: ...prebiotically plausible phosphorylating agents

Reviewer: Prebiotic plausibility constitutes a scientific claim that needs to be motivated and whose premises need to be stipulated. In the present version of the article we find no demonstration of this claim, nor references to the literature where such a claim is voiced.

Although the rigor behind this terminology has undergone considerable erosion in the last 50 years, members of the community have started to voice concerns (see: Benner SA. Prebiotic plausibility and networks of paradox-resolving independent models. *Nat. Commun.* 2018 Dec 12;9(1):5173. doi: 10.1038/s41467-018-07274-y). One should not freely call a compound ‘prebiotically plausible’, it needs to be motivated.

In the case of DAP and MAP, there is indeed a recent body of literature that explores and argues for these species to be prebiotically plausible phosphorylating agents, for instance: Diamidophosphate (DAP) – A Plausible Prebiotic Phosphorylating Reagent with a Chem to BioChem Potential? Osumah, A.; Krishnamurthy, R. *ChemBioChem*, 2021, 22, 3001-3009. Geochemical Sources and Availability of Amidophosphates on the Early Earth. Gibard, C.; Jiménez, E. I.; Gorrell, I. B.; Kee, T. P.; Pasek, M. A.; Krishnamurthy, R. *Angew. Chemie Int. Ed.* 2019, 58, 8151-8155.

Authors: The reaction cycle is enzyme-free and produces a phosphorylated amino acid or peptide that hydrolyzes without requiring enzymes with a tunable half-life.

Reviewer: This phrase can be improved, for instance by adding ‘and’ in front of ‘with a tunable half-life’.

Results and discussion

Exploration of the chemical reaction network.

Reviewer: Some minor mistakes.

Prebiotically -> prebiotically

The kinetic model is unusually predictive for a system of this level of complexity, especially for the species 3-pH. Nevertheless, there is unexplained variation, i.e. qualitative and quantitative deviations for the other species. It would be instructive if the authors could include a short speculation on the possible sources of this variation.

The labels C, D, F are missing from Fig.3.

Conclusions

Reviewer: The conclusion is rather short, mentions few results and is not very specific. I recommend expanding the conclusion to be more reflective of the content of the paper.

References

Reviewer: There is an error in the author list for reference 7.

Sup. Matt.

Reviewer: Among the most valuable products of this research is the high-quality kinetic data, involving multiple species, many timepoints, and repetitions. If this was not already being done, I encourage the authors to make it as easy as possible for their colleagues to access the raw data, e.g. by storing it in online repositories.

Reviewer #3 (Remarks to the Author):

This manuscript describes the use of histidine phosphorylation by amidophosphates as a new reaction network for nonequilibrium systems chemistry. The authors establish the fundamental chemistry by optimizing the conditions and demonstrate its applicability through the formation of transient coacervate droplets.

The design of this system is simple and very elegant, and in my view is likely to be of broad interest to the field, especially considering some relevance to prebiotic chemistry. The authors have been thorough in exploring the parameter space for this reaction. I support publication in Nature Communications subject to a few comments.

1. The authors make use of a large, complex kinetic model. My major concern is the possibility that some of the parameters determined by fitting may be correlated (i.e., the model cannot measure them accurately because simultaneous changes in two or more cancel out in the concentration vs time simulations). If this is the case, the reported parameters (Table S1) may not be all that meaningful. There are a few things I noticed that

seem like red flags:

- In some cases (e.g., k_3 for Ac-GHG-OH vs acY'DHDDam) there seem to be very large discrepancies in the k values for what would seem to be similar reactions.
- In other cases, the errors determined (I assume) from the regression appear to be very large relative to the k 's themselves (e.g., k_5 , k_7).
- In one case $k_4 \ll k_1$, which seems chemically unreasonable.

This issue should be discussed in the SI. If it cannot be adequately addressed, I would suggest removing explicit discussion of any parameters that are not determined by the data. It may only be possible to state that the mechanism is consistent with the data but that specific parameters cannot be accurately determined. To my mind this would not substantially affect interest in this work.

2. On a related note, how were the errors determined on the parameters in Tables S1 and S2? Is this through multiple replicates, or are they errors from the fits? This should be specified.

3. Some authors, notably Astumian, Aprahamian, and Goldup, have objected to how some terminology is used in this field and how some basic concepts are frequently described. I don't personally take as extreme a view, but I do think there is merit to these arguments. The manuscript describes unstable fuel molecules decomposing to waste, which I believe is one of the explanations that is often objected to. Really the issue is not specific unstable molecules, but rather that the reaction network must be driven by a reaction where the reactant and product concentrations are out of equilibrium; the distinction between fuel and waste is essentially arbitrary. That is, ADP can be a fuel for a reaction network if its concentration is above its equilibrium concentration. I would suggest making the language used in the introduction of this manuscript a bit more rigorous.

4. I'm a bit confused by how scientific notation is being used throughout the SI. In Table S1, there are frequent examples of numbers and uncertainties with both parts expressed in scientific notation (e.g., $1.40e-3 \pm 7.58e-5$). In Table S2, however, it seems like the exponent

of the uncertainty is supposed to apply to both parts (e.g., $2.34 \pm 0.11e-1 = 2.34e-1 \pm 0.11e-1$?). I'm therefore not sure what convention some numbers are following (e.g., $2.17 \pm 0.50e-3$ in Table S1). This should be clarified. I would personally suggest making it very explicit throughout, such as " $(2.34 \pm 0.11) \times 10^{-1}$ ".

5. There are some incorrect table numberings in the SI. For example, there are frequent references to Table S7 that I think are supposed to be to Table S1. These should all be checked.

6. The notation is a bit confusing. For example, "3-pHis" (text) and "3-pH" (figures) appear to be used interchangeably.

7. I may be misunderstanding, but I believe the discussion on lines 113–115 is saying that the half life for loss of phosphate from 1,3-bpHis is the same or slightly longer than that for 1-pHis. This seems unlikely? If there are correlated parameters as noted above, I am concerned about quantitative assessments like this (depending on what's correlated with what, of course).

8. A bunch of the labels (C, D, F) are missing in Figure 3.

9. I noticed that the authors have tended to use near-stoichiometric or sub-stoichiometric ratios of MAP to peptide. Is there a reason for this? Especially considering that the system is not tremendously efficient (Figure 2F/G), it would seem like excess fuel might be useful.

Response to the reviewers for: NCOMMS-23-62971

Reviewer #1 (Remarks to the Author):

Key results: The authors describe a strategy to transiently phosphorylate histidine, with the goal of mimicking the transient activation of enzymes in cells. Monoamidophosphate (MAP) is the main phosphorylating agent, and water is the dephosphorylating agent. Histidine catalyses the hydrolysis of MAP, getting phosphorylated/de-phosphorylated in the process. When histidine is embedded in a tripeptide, de-phosphorylation proves to be too slow for the reaction cycle to be considered dynamic, and the authors deploy pyridine to accelerate it. In the presence of MAP and pyridine, the half-life of the phosphorylated peptide is reduced to a few hours. Experimental data and kinetic modeling are used to depict the flow of phosphate in the system. The authors show that when the reaction is carried out in the presence of a polycation, supramolecular assemblies (coacervates) form and disappear following the dynamic phosphorylation of histidine.

Authors: We thank the reviewer for the time invested in the manuscript. We addressed the comments below.

Validity & Significance: The experimental data in the manuscript is complete and the kinetic model provides a good fit. However, I believe the concept and strategy described in this manuscript are not significantly novel, and would not have a sufficient impact in the community to grant publication in Nature Communications. Transient compartmentalization controlled by phosphorylation has been shown previously (DOI: 10.1038/nchem.2414, 10.1039/c7sm01897e). It is true that the manuscript provides a non-enzymatic version, but still the applicability is limited by the fact that the reaction only works with the pair MAP/histidine (tables in Figure 1). We have now added the reference of Keating and Huck's work. Moreover we also cited Spruijt's work describing ATP based active droplets.

Recent literature on prebiotic phosphorylation tends to look for agents with a broader spectrum (e.g. DOI 10.1021/jacs.3c08539, 10.1038/s41467-021-25555-x); or in the case of histidine itself, for functional reactions with greater resemblance to natural catalytic cycles (see alternative with imidazole phosphate in Maguire ChemRXiv 2023). For the reaction cycle to operate like in biology, both activation and deactivation occur on similar timescales without side reactions. In this work, we focused on a quantitative understanding of a chemical reaction cyclus rather than its breadth. While we are convinced there are other substrate and fuel combinations, besides the ones that we tried, that would also work, we

decided to focus on the combination of histidine and its peptide derivatives with MAP as a fuel.

MAP, as the authors point out themselves, is not the most prebiotically plausible phosphorylating agent; and the use of pyridine as a substrate (that is mostly phosphorylated directly and competes with histidine) is somewhat confusing and undermines, in my opinion, its potential impact in the prebiotic chemistry field.

We appreciate the input, and we toned down the claim regarding prebiotic relevance.

Moreover, in line with reviewer two, we cited recent work on the prebiotic relevance of MAP.

The results on the formation of compartments are also weakened by the use of PEG 8k (at 2.5% concentration), a crowding agent known to promote and interfere with phase separation – more than a simple turbidity enhancer. Combined with the fact that most micrographs show only one droplet, or one microreactor, it indicates that the supramolecular results may be not robust or reproducible, and that the impact on the protocell/dynamic assemblies community would be very limited.

The way the data was presented indeed only showed one droplet per micrograph. We now added further evidence of the droplet formation. On the one hand, we show the formation of multiple droplets within one microreactor without the need of PEG. On the other hand, we show multiple microreactors containing droplets within one micrograph. Finally, the evolution of the droplet volume over time was performed in triplicate demonstrating that it is reproducible.

I include below some specific minor and major concerns about the manuscript.

Clarity and context:

1. The authors use terms such as "active droplets" in the title, "self-sustaining complex coacervate droplets", but do not justify it in the text or in the results. Active droplets would imply a non-equilibrium behavior, which is not clear to me in the data (the droplets emerge, coalesce, and dissolve). Self-sustaining would imply that the droplets facilitate the phosphorylation reaction that leads to their own formation, which was not probed in the manuscript.

We agree that self-sustaining would imply a more complex system that can, for example, facilitate its own formation. To clarify and not confuse our readers, we've changed from self-sustaining to active, with active meaning with activated precursor/product present (highlighted in yellow).

Our work demonstrates the emergence, the coalescence, and the dissolution of droplets, which can only happen out of equilibrium.

2. The description of the reaction as a cycle, or as a network (lines 64, 129), may be misleading, with the deactivation agent being the reaction's solvent (or pyridine), and the fuel MAP not being regenerated.

This was not meant to be misleading. We picture it as a cycle from the histidine's point of view. The histidine in our cycle undergoes cyclical phosphorylation drive by the hydrolysis of phosphorylating agent. Conceptually, this is similar to how proteins can undergo cyclical phosphorylation and dephosphorylation driven by the hydrolysis of ATP. We've now further clarified in the main text (highlighted in yellow).

3. I am missing the explanation as to how short of a half-life is desirable in a biological or chemical context, and why the choice to rely on water or pyridine as de-activating agents, as opposed of some prebiotically relevant substrates. The authors could include a more specific discussion on histidyl kinases, instead of the general introduction on ATP.

We added a part on the histidine kinases to the introduction (highlighted in yellow). We were interested in what influences the half-lives.

4. Figures: scheme 1 would be more appropriate as a graphical abstract, it is taking up quite a lot of space but not adding as much information. Figure 1 could be reorganized to take up less space (the table format may not be necessary).

We agree scheme 1 is more like a graphical abstract, and we reduced the size and compressed it. Nevertheless, it includes all the information needed (histidine-based precursor as 5membered ring, phosphorylation coupled to droplet formation with information about the constitution of the droplets). Besides nature communications does not work with graphical abstract, which is why we use Scheme 1 to graphically demonstrate the idea of the work.

In our opinion, the table format in Fig. 1 is in this case useful, nicely aligned and gives a good overview.

Figure 2 contains some redundant data, panels F and G could be in supplementary information. Instead, a table for half-life and conversion% would be helpful.

In our opinion, the panels (bar graphs) are clearer than tables. In a table the conversion% is less obvious than the bar graph with colorfully highlighted %ratio, as well as the change of half-lives under different conditions. But if you want to directly compare the numbers, we added the half-lives to Table S1 and S3 in the SI.

In Figure 3, what does Y'Dam mean?

We've changed it to peptide 1.

5. The clarity of the manuscript could be increased by changing the acronyms. Instead of 1-pH, His1-P for example, as pH already has its own meaning.

We appreciate the feedback. It is also in agreement with the literature, so we changed it to x-pHis.

6. Line 27: The sentence starting with "Does not occur..." is incomplete. The text from line 27 up to 39 needs to be restructured, it is confusing and not conveying its message – which I think is that biological cycles benefit from using ATP as a fuel due to it being thermodynamically unstable but kinetically stable.

We rephrased and clarified the sentences.

7. Line 187: there is a typo, in "Either it is hydrolyzed directly or through phosphorylating histidine (unwanted)" it should be pyridine, not histidine.

In line 192 there is a repetition "activation phosphorylation". This paragraph could be made clearer and the quantitative data mentioned could be included in a table elsewhere for clarity.

Thank you, we rephrased the sentence.

Data and methodology:

8. Both experimental and modeling kinetic data are well presented, error bars are shown but it is not mentioned if the mean is taken over multiple experiments or different measurements or time points. I appreciate how the methodology was extensively described.

We appreciate the comment. In line with nature communication policy, we updated the presentation of the experimental data. We now show each data point of each experiment.

The only data with mean error bars is the droplet volume/microreactor volume (droplets analyzed per micrograph > 10), but the triplicates are plotted individually.

9. Overall, some experiments could be better motivated in the text: for example, it is not clear to me that such a complete kinetic modeling is necessary, or how it can contribute to the understanding of the system.

We performed the kinetic modeling to predict the concentration evolution of the different species as precisely as possible. We also want to help others start a project based on this system. We have clarified in the main text.

Another example is why the authors choose to conduct microscopy in microfluidically-produced micro-reactors, and not on a typical microscopy slide/chamber - what advantages to the paper's goals does it bring?

We address this comment in point 10.

The ITC and DNA partitioning data is included but not well integrated within the text - how is this data relevant to the manuscript?

We integrated it better in the text.

10. The microscopy images do not immediately point to coacervate droplets (if you compare to other peptide/peptide complex coacervates in related literature), although I understand that FRAP supports it. Perhaps resolution and visibility would be increased without the microreactors. And instead of adding PEG, I would consider reducing buffer concentration.

Thank you for your comment. We have now added data that shows that the droplets fuse. Besides, we have clarified that we have used the dye sulforhodamine B, a dye that is water soluble demonstrating an aqueous interior.

We used microreactors to analyze the droplets because the droplets do not adhere to the microreactor walls. In contrast, they quickly adhere to glass slides. We clarified this in the main text (highlighted in yellow).

We use PEG instead of reducing the buffer concentration to keep the ionic strength the same as in the microreactors.

11. Scale bars are often missing.

Thank you for your comment we added them, where needed.

Reviewer #2 (Remarks to the Author):

Referee report

Active droplets through enzyme-free, dynamic phosphorylation

Authors: Simone Poprawa,¹ Michele Stasi,¹ Monika Wenisch,¹ Brigitte A. K. Kriebisch,¹ Judit Sastre,¹ Job Boekhoven¹

Reviewer: The article presents an elegant system for the construction of active droplets using dynamic phosphorylation in an enzyme-free manner, and hence furnishes a platform that others can readily built upon. This work will likely be picked up rapidly by several communities (for instance systems chemistry, active matter, and origins of life) and can be expected to become a highly cited paper. As such, I am overall favorable to its publication in Nature Communications. Nevertheless, in terms of writing, presentation and contextualization, the present version feels unfinished and the scholarly quality of the work can be favorably improved.

Thank you for your time invested in the manuscript. We have addressed your comments below.

In chronological order:

Abstract

Reviewer: The abstract provides hardly more information than the title about the content of the paper and remains vague in instances where being precise would hardly consume more space, for instance it may be pointed out that the system is specific to histidine, and what phosphorylating agent is used.

We rephrased the sentences.

Authors: Enzyme-free, synthetic systems that use dynamic phosphorylation to regulate supramolecular processes do not exist

Reviewer: Do not exist is a confusing phrasing. The authors presumably mean that such systems 'have not yet been reported, to the best of our knowledge'.

Thank you, we rephrased the sentences.

Authors: Our new reaction cycle will be valuable as a model for biological phosphorylation but can also offer insights into protocell formation.

Reviewer: Given the long history of MAP and DAP chemistry and renewed recent interest (which should be cited), the possessive phrasing 'our new reaction cycle' feels unnecessary here. The article does not clarify in what sense(s) the reaction cycle may be expected to be deliver on these promises. Since the phosphorylation is specific to histidine, it is not a priori obvious that it provides a model for biological phosphorylation.

Indeed, “new” is too possessive. We added a part of histidine phosphorylation in the introduction (highlighted in yellow).

Introduction

Reviewer: The introduction explains at length certain aspects that are widely known, while assuming intimate familiarity of the reader with communities and highly specific branches of literature that are not cited. In particular, pertinent recent literature involving phosphates in systems chemistry and MAP/DAP in prebiotic chemistry appears to be overlooked.

With respect to the former, notably:

Spontaneous and Selective Peptide Elongation in Water Driven by Aminoacyl Phosphate Esters and Phase Changes, Kun Dai, Mahesh D. Pol, Lenard Saile, Arti Sharma, Bin Liu, Ralf Thomann, Johanna L. Trefs, Danye Qiu, Sandra Moser, Stefan Wiesler, Bizan N. Balzer, Thorsten Hugel, Henning J. Jessen, and Charalampos G. Pappas*, J. Am. Chem. Soc. 2023, 145, 48, 26086–26094, <https://doi.org/10.1021/jacs.3c07918>

A Ribonucleotide <-> Phosphoramidate Reaction Network Optimized by Computer-Aided Design. Andreas Englert, Julian F. Vogel, Tim Bergner, Jessica Loske, and Max von Delius. J. Am. Chem. Soc. 2022, 144, 33, 15266–15274

We added the references.

Authors: Does not occur on a reasonable time scale;

Reviewer: This phrase is grammatically incorrect.

We rephrased the sentence.

Authors: Their need for highly evolved enzymes also offers limited insight into phosphorylation before life existed

Reviewer: This statement is suggestive and lacks nuance. It suggests an implicit historical claim that phosphorylation is a process that predates life. This is a popular – though unproven - conjecture in a niche of origins of life, where it is more broadly believed that modern biomolecules coincide with those present at abiogenesis. Since much of the prospective readership is not directly from the origins community and may not be aware of niche conjectures, any reference to them needs to be contextualized. If the authors choose to do so, they may also want to cite appropriate literature that lays out these conjectures and viewpoints thereon. A recent work on conjectured roles of phosphate groups in prebiotic chemistry is for instance: Ring Opening of Glycerol Cyclic Phosphates Leads to a Diverse Array of Potentially Prebiotic Phospholipids, Maiia Aleksandrova, Fidan Rahmatova, David A. Russell, and Claudia Bonfio. J. Am. Chem. Soc. 2023, 145, 47, 25614–25620, <https://doi.org/10.1021/jacs.3c07319>

It should be stressed that biochemistry-first as a conjecture has been questioned many times over, even by its proponents within the origins literature, a relatively recent source being:

Life's Biological Chemistry: A Destiny or Destination Starting from Prebiotic Chemistry? R. Krishnamurthy, Chem. Eur. J. 2018, 24, 16708.

A more nuanced phrasing would be ‘.. into the roles phosphorylation may potentially have played before life existed’.

We totally agree and appreciate the thoughtful feedback. We nuanced the statement and added supporting citations.

Authors: ...prebiotically plausible phosphorylating agents

Reviewer: Prebiotic plausibility constitutes a scientific claim that needs to be motivated and whose premises need to be stipulated. In the present version of the article we find no demonstration of this claim, nor references to the literature where such a claim is voiced.

Although the rigor behind this terminology has undergone considerable erosion in the last 50 years, members of the community have started to voice concerns (see: Benner SA. Prebiotic plausibility and networks of paradox-resolving independent models. Nat. Commun. 2018 Dec 12;9(1):5173. doi: 10.1038/s41467-018-07274-y). One should not freely call a compound ‘prebiotically plausible’, it needs to be motivated.

In the case of DAP and MAP, there is indeed a recent body of literature that explores and argues for these species to be prebiotically plausible phosphorylating agents, for instance: Diamidophosphate (DAP) – A Plausible Prebiotic Phosphorylating Reagent with a Chem to BioChem Potential? Osumah, A.; Krishnamurthy, R. ChemBioChem, 2021, 22, 3001-3009. Geochemical Sources and Availability of Amidophosphates on the Early Earth. Gibard, C.; Jiménez, E. I.; Gorrell, I. B.; Kee, T. P.; Pasek, M. A.; Krishnamurthy, R. Angew. Chemie Int. Ed. 2019, 58, 8151-8155.

Thank you for reminding us, of course these are very important publications and need to be cited.

Authors: The reaction cycle is enzyme-free and produces a phosphorylated amino acid or peptide that hydrolyzes without requiring enzymes with a tunable half-life.

Reviewer: This phrase can be improved, for instance by adding ‘and’ in front of ‘with a tunable half-life’.

We changed the sentence.

Results and discussion

Exploration of the chemical reaction network.

Reviewer: Some minor mistakes.

Prebiotically -> prebiotically

We corrected the typo.

The kinetic model is unusually predictive for a system of this level of complexity, especially for the species 3-pH. Nevertheless, there is unexplained variation, i.e. qualitative and quantitative deviations for the other species. It would be instructive if the authors could include a short speculation on the possible sources of this variation.

The reaction rate constant of the 3-pHis was determined empirically when no other species or fuel were present anymore. Therefore, the predicted and the experimentally measured evolution fit nicely. The hydrolysis of MAP was also calculated empirically, but the other constants (up to 17) were fitted and thus show higher deviations (Table S1, error of the fit) as there are many combinations possible, also with correlated parameters. Despite that, the fits align very well with the experimental data.

As you and reviewer 3 mentioned, the kinetic model is very good for predicting the evolution of a single species and can potentially help to design a self-assembling system. However, the reaction rate constants, which are not determined by the data, should not be compared or addressed directly. We have removed any references to half-lives or rate constants that were determined by the kinetic model.

Besides, we have added the comment on the value and the limitation of the kinetic model in the SI.

The labels C, D, F are missing from Fig.3.

We changed the presentation of our data and also added the labels.

Conclusions

Reviewer: The conclusion is rather short, mentions few results and is not very specific. I recommend expanding the conclusion to be more reflective of the content of the paper.

We have addressed your comments on the conclusion.

References

Reviewer: There is an error in the author list for reference 7.

We updated the references.

Sup. Matt.

Reviewer: Among the most valuable products of this research is the high-quality kinetic data, involving multiple species, many timepoints, and repetitions. If this was not already being done, I encourage the authors to make it as easy as possible for their colleagues to access the raw data, e.g. by storing it in online repositories.

Thank you, we appreciate that! We added the Excel sheet with the raw data of the kinetic experiments and the COPASI sample file.

Reviewer #3 (Remarks to the Author):

This manuscript describes the use of histidine phosphorylation by amidophosphates as a new reaction network for nonequilibrium systems chemistry. The authors establish the fundamental chemistry by optimizing the conditions and demonstrate its applicability through the formation of transient coacervate droplets.

The design of this system is simple and very elegant, and in my view is likely to be of broad interest to the field, especially considering some relevance to prebiotic chemistry. The authors have been thorough in exploring the parameter space for this reaction. I support publication in Nature Communications subject to a few comments.

Thank you for your positive feedback, we appreciate your time invested in the manuscript.

1. The authors make use of a large, complex kinetic model. My major concern is the possibility that some of the parameters determined by fitting may be correlated (i.e., the model cannot measure them accurately because simultaneous changes in two or more cancel out in the concentration vs time simulations). If this is the case, the reported parameters (Table S1) may not be all that meaningful. There are a few things I noticed that seem like red flags:

- In some cases (e.g., k_3 for Ac-GHG-OH vs acY'DHDDam) there seem to be very large discrepancies in the k values for what would seem to be similar reactions.
- In other cases, the errors determined (I assume) from the regression appear to be very large relative to the k 's themselves (e.g., k_5 , k_7).
- In one case $k_4 \ll k_1$, which seems chemically unreasonable.

This issue should be discussed in the SI. If it cannot be adequately addressed, I would suggest removing explicit discussion of any parameters that are not determined by the data. It may only be possible to state that the mechanism is consistent with the data but that specific parameters cannot be accurately determined. To my mind this would not substantially affect interest in this work.

Thank you for your detailed comment. We have distanced ourselves to compare the fitted values directly, but use them as a guideline and compare empirically determined parameters. Please see the answer to reviewer 2.

2. On a related note, how were the errors determined on the parameters in Tables S1 and S2? Is this through multiple replicates, or are they errors from the fits? This should be

specified.

We added the explanation to the description of the tables.

3. Some authors, notably Astumian, Aprahamian, and Goldup, have objected to how some terminology is used in this field and how some basic concepts are frequently described. I don't personally take as extreme a view, but I do think there is merit to these arguments. The manuscript describes unstable fuel molecules decomposing to waste, which I believe is one of the explanations that is often objected to. Really the issue is not specific unstable molecules, but rather that the reaction network must be driven by a reaction where the reactant and product concentrations are out of equilibrium; the distinction between fuel and waste is essentially arbitrary. That is, ADP can be a fuel for a reaction network if its concentration is above its equilibrium concentration. I would suggest making the language used in the introduction of this manuscript a bit more rigorous.

We have adjusted the introduction to define fuel better.

4. I'm a bit confused by how scientific notation is being used throughout the SI. In Table S1, there are frequent examples of numbers and uncertainties with both parts expressed in scientific notation (e.g., $1.40e-3 \pm 7.58e-5$). In Table S2, however, it seems like the exponent of the uncertainty is supposed to apply to both parts (e.g., $2.34 \pm 0.11e-1 = 2.34e-1 \pm 0.11e-1$?). I'm therefore not sure what convention some numbers are following (e.g., $2.17 \pm 0.50e-3$ in Table S1). This should be clarified. I would personally suggest making it very explicit throughout, such as $(2.34 \pm 0.11) \times 10^{-1}$.

Thank you, we changed it.

5. There are some incorrect table numberings in the SI. For example, there are frequent references to Table S7 that I think are supposed to be to Table S1. These should all be checked.

We updated the numbering.

6. The notation is a bit confusing. For example, "3-pHis" (text) and "3-pH" (figures) appear to be used interchangeably.

To be consistent, also with literature, we changed to x-pHis.

7. I may be misunderstanding, but I believe the discussion on lines 113–115 is saying that the half life for loss of phosphate from 1,3-bpHis is the same or slightly longer than that for 1-pHis. This seems unlikely? If there are correlated parameters as noted above, I am concerned about quantitative assessments like this (depending on what's correlated with what, of course).

See the answer to your comment 1.

8. A bunch of the labels (C, D, F) are missing in Figure 3.

We changed presentation of our data and also added the labels.

9. I noticed that the authors have tended to use near-stoichiometric or sub-stoichiometric ratios of MAP to peptide. Is there a reason for this? Especially considering that the system is not tremendously efficient (Figure 2F/G), it would seem like excess fuel might be useful.

We agree that the yield would be higher with a higher amount of fuel.

However, an excess of fuel also elongates the total experimental time further. To keep the experiment on a reasonable timescale, we usually work with stoichiometric amounts.

Moreover, to potentially observe the bisphosphorylated species, we used nearly stoichiometric and high amount of peptide to fuel.

REVIEWER COMMENTS

Reviewer #1 (Remarks to the Author):

The authors addressed some of my concerns, but my major concerns regarding the publication of this manuscript in Nature Communications remain. That said, at least the writing needs further improvement in order for me to be comfortable supporting publication. I really think the manuscript will benefit from a more rigorous writing, and that time put on these final changes will be well spent.

I refer to the numbering in my previous report:

1. Partially addressed. In line 229, I disagree from the definition of active droplets or active protocells (also comparing to David Zwicker's group definition): In this context, active means that the droplets are sustained as long as a certain amount of product is present. The way it is written at least, it is not a remarkable behaviour – that the droplets exist while their building blocks exist. Reaction-controlled coacervates are not an active per se; it helps to make your case if you compare these dynamic droplets, sustained by the reaction, to droplets formed from building blocks that don't form or decay in situ.

2. Despite the explanation in the reply to my comments, the introduction text still lacks specificity. As a reader, I am still not convinced that this is a cycle, or a controlled cycle: surely, while there is fuel, there is phosphorylation; and surely when you stop adding fuel, hydrolysis dominates, but that does not mean control. A key point of a cycle would be to demonstrate that His phosphorylation (and coacervates) can be recovered by a fresh addition of MAP. Can you show this?

a. Line 27 is an example of what I mean by lack of specificity in the writing: That way, the equilibrium position for the fuel-to-waste equilibrium lies to the right such that most fuel will eventually be converted into waste. (and btw line 33 can start a new paragraph)

b. In line 287, you write "reaction cycle that phosphorylates histidine as an amino acid or in a peptide at the expense of phosphorylating agents". There is no need to say "at the expense of phosphorylating agents", this is the case in any chemical reaction. This and other similar repetitions take space from a more meaningful discussion/conclusion.

3. Addressed.

4. Since you chose to keep the table in Figure 1, I find it that there needs to be some discussion in the text as to why the cycle only works with histidine, or why, if we know from the literature that MAP phosphorylates imidazole groups, other residues were also tested. What was the hypothesis?

5. Addressed.

6. Addressed.

7. Addressed.

8. Addressed.

9. Partially addressed. The use of microreactors still confuses me, as there are many ways to passivate glass slides to prevent droplet adhesion. Can you at least include in SI micrographs of droplets outside these microreactors? This would show that the microreactor itself is not crucial to the droplets' properties.

10. In SI Figure 10, I see you now added the whole field of view, showing all microreactors. Can you include the emission channel as well, which is what you mostly show in the main text? The caption says it shows the emergence, growth and dissolution of the droplets, but I can only see emergence and dissolution.

11. Addressed.

New comments:

12. Line 161: consider reorganizing the long list of references. The comparison to the literature is barely made, yet there are 11 references listed. I prefer it if you elaborate on this discussion rather than remove references, as I think it will make the manuscript stronger.

13. The conclusions are still vague and imprecise despite another reviewer's comment. Most statements are based on this system being a dynamic cycle and the droplets being active, so

I think resolving these conceptual problems in the main text will help better support the conclusions.

Reviewer #2 (Remarks to the Author):

All reviewer points are addressed well.

Reviewer #3 (Remarks to the Author):

In my opinion, the authors have addressed the points I raised in my original review. I believe that the manuscript is now suitable for Nature Communications.

REVIEWER COMMENTS

Reviewer #1 (Remarks to the Author):

The authors addressed some of my concerns, but my major concerns regarding the publication of this manuscript in Nature Communications remain. That said, at least the writing needs further improvement in order for me to be comfortable supporting publication. I really think the manuscript will benefit from a more rigorous writing, and that time put on these final changes will be well spent.

I refer to the numbering in my previous report:

1. Partially addressed. In line 229, I disagree from the definition of active droplets or active protocells (also comparing to David Zwicker's group definition): In this context, active means that the droplets are sustained as long as a certain amount of product is present. The way it is written at least, it is not a remarkable behaviour – that the droplets exist while their building blocks exist. Reaction-controlled coacervates are not an active per se; it helps to make your case if you compare these dynamic droplets, sustained by the reaction, to droplets formed from building blocks that don't form or decay in situ.

We agree and changed it for:

In this context, active means that the droplet material is controlled by two chemical reactions: activation and deactivation.

We used the nomenclature as introduced by Zwicker: „Droplets can become chemically active if the material of the droplet is produced and destroyed by chemical reactions. An example that resembles a simple protocell is shown schematically in Fig. 1a. The droplet is formed by a droplet material D that is generated inside the droplet from a high-energy precursor N, which plays the role of a nutrient. Droplet material can degrade into a lower energy component, W, that plays the role of a waste, which leaves the droplet by diffusion. The droplet can survive if N is continuously supplied and W is continuously removed. This can be achieved by recycling N using an external energy source such as a fuel or radiation.“¹

(1) Zwicker, D.; Seyboldt, R.; Weber, C. A.; Hyman, A. A.; Jülicher, F. Growth and division of active droplets provides a model for protocells. *Nature Physics* **2017**, *13* (4), 408-413. DOI: 10.1038/nphys3984.

2. Despite the explanation in the reply to my comments, the introduction text still lacks specificity. As a reader, I am still not convinced that this is a cycle, or a controlled cycle: surely, while there is fuel, there is phosphorylation; and surely when you stop adding fuel, hydrolysis dominates, but that does not mean control. A key point of a cycle would be to demonstrate that His phosphorylation (and coacervates) can be recovered by a fresh addition of MAP. Can you show this?

We tested refueling with three cycles on the amino acid His on the NMR, to demonstrate that we can recover the phosphorylation by fresh addition of MAP. Besides, we also refueled the peptide with droplets two times on the plate reader. The graphs can be found in the SI.

a. Line 27 is an example of what I mean by lack of specificity in the writing: That way, the equilibrium position for the fuel-to-waste equilibrium lies to the right such that most fuel will eventually be converted into waste. (and btw line 33 can start a new paragraph)

We rephrased to: That way, the equilibrium position for the fuel-to-waste equilibrium lies on the waste side such that most fuel will eventually be converted into waste. This fuel-to-waste conversion should be slow so that a catalyst can accelerate it. Put differently, the fuel should be thermodynamically unstable, such as kinetically inert. In the catalytic reaction cycle, the catalyst accelerates the fuel-to-waste conversion. In doing so, the catalyst is temporarily activated by reacting with the fuel, after which it spontaneously deactivates. In other words, the catalyst can undergo numerous activation-deactivation cycles, and the fuel is converted fast due to the presence of the catalyst.

Highlighted in pink.

b. In line 287, you write “reaction cycle that phosphorylates histidine as an amino acid or in a peptide at the expense of phosphorylating agents”. There is no need to say "at the expense of phosphorylating agents", this is the case in any chemical reaction. This and other similar repetitions take space from a more meaningful discussion/conclusion.

We agree that it is not needed. We removed it where we thought it was not required. Highlighted in pink.

3. Addressed.

4. Since you chose to keep the table in Figure 1, I find it that there needs to be some discussion in the text as to why the cycle only works with histidine, or why, if we know from the literature that MAP phosphorylates imidazole groups, other residues were also tested. What was the hypothesis?

We focused on amino acids and their peptides as they are powerful building blocks in peptide self-assembly, including liquid-liquid phase separation, which we show later in this paper. We emphasized our reasoning in the text and highlighted it in pink.

5. Addressed.

6. Addressed.

7. Addressed.

8. Addressed.

9. Partially addressed. The use of microreactors still confuses me, as there are many ways to passivate glass slides to prevent droplet adhesion. Can you at least include in SI micrographs of droplets outside these microreactors? This would show that the microreactor itself is not crucial to the droplets' properties.

To clarify, the microreactor helps us to study droplet formation without the downsides of flow in the sample, droplets settling on the glass, and droplets escaping out of the focal plane during imaging. We also understand your concern, so we added micrographs of droplets outside the microreactors to the SI.

10. In SI Figure 10, I see you now added the whole field of view, showing all microreactors. Can you include the emission channel as well, which is what you mostly show in the main text? The caption says it shows the emergence, growth and dissolution of the droplets, but I can only see emergence and dissolution.

We added the emission channel of sulforhodamine B and the micrographs after 9 and 25 h. Fig. 3 f shows the growth more clearly as the percentage of droplet material per reactor correlates with the droplet's size.

11. Addressed.

New comments:

12. Line 161: consider reorganizing the long list of references. The comparison to the literature is barely made, yet there are 11 references listed. I prefer it if you elaborate on this discussion rather than remove references, as I think it will make the manuscript stronger.

Indeed, 11 references were a lot, and some of them fitted better to other statements that have been made before. Changes are highlighted in pink.

13. The conclusions are still vague and imprecise despite another reviewer's comment. Most statements are based on this system being a dynamic cycle and the droplets being active, so I think resolving these conceptual problems in the main text will help better support the conclusions.

We have now clarified what we mean by active droplets (following your earlier statement) and addressed the fact that it is a cycle (following your proposed experiments). So, the conceptual problems should now be resolved.

REVIEWERS' COMMENTS

Reviewer #1 (Remarks to the Author):

The authors have now addressed all remarks and included new experiments. I believe the manuscript's clarity and scientific rigor improved greatly during the revision process, and appreciate their effort in doing so. I support publication of the most recent version.